# CONTRACTIVE DIFFUSION POLICIES

**Amin Abyaneh**[1,3]**, Charlotte Morissette**[2,3]**, Mohamad H. Danesh**[2,3]**, Anas El Houssaini**[2,3]**,
David Meger**[2,3]**, Gregory Dudek**[2,3]**, Hsiu-Chin Lin**[1,2,3]

[1]Department of Electrical and Computer Engineering, McGill University
[2]Department of Computer Science, McGill University
[3]Mila–Quebec AI Institute          **Correspondence:** amin.abyaneh@mail.mcgill.ca

## ABSTRACT

Diffusion policies have emerged as powerful generative models for offline pol-
icy learning, whose sampling process can be rigorously characterized by a score
function guiding a stochastic differential equation (SDE). However, the same score-
based SDE modeling that grants diffusion policies the flexibility to learn diverse
behavior also incurs solver and score-matching errors, large data requirements, and
inconsistencies in action generation. While less critical in image generation, these
inaccuracies compound and lead to failure in continuous control settings. We intro-
duce **C**ontractive **D**iffusion **P**olicies (CDPs) to induce contractive behavior in the
diffusion sampling dynamics. Contraction pulls nearby flows closer to enhance ro-
bustness against solver and score-matching errors while reducing unwanted action
variance. We develop an in-depth theoretical analysis along with a practical imple-
mentation recipe to incorporate CDPs into existing diffusion policy architectures
with minimal modification and computational cost. We evaluate CDPs for offline
learning by conducting extensive experiments in simulation and real world settings.
Across benchmarks, CDPs often outperform baseline policies, with pronounced
benefits under data scarcity. **Project page**: contractive-diffusion.github.io

## 1 INTRODUCTION

Diffusion policies have substantially advanced offline policy learning, particularly in robotics and
control, where they are trained with reinforcement learning (RL) or imitation learning (IL) (Chi et al.,
2023; Ke et al., 2024; Huang et al., 2024; Wang et al., 2024; Sridhar et al., 2024). These improvements
stem from stable training and strong representational capabilities of conditional diffusion models
in capturing long-horizon and multimodal behavior (Sohl-Dickstein et al., 2015; Ho et al., 2020).
Essentially, diffusion models corrupt actions with noise and learn a score function that, conditioned
on the state, iteratively denoises the corrupted actions to reconstruct the underlying conditional action
distribution. This corruption-denoising process can be formally described by a stochastic differential
equation (SDE), where actions are obtained through differential equation solvers (Song et al., 2021b).

Despite recent advances, diffusion policies still demand sub-
stantial samples to learn accurate score functions and are vul-
nerable to solver errors. Iterative diffusion sampling causes
inaccuracies in (i) score estimation, (ii) discretization, and
(iii) numerical integration to accumulate across denoising
steps (Zheng et al., 2023; Tang & Zhao, 2024). In practice,
the sampling process also suffers from inconsistency in action
generation given the same state (Gao et al., 2025). Unlike
image generation, inaccuracies of this sort are detrimental in
robotics and control, as even small deviations from the data
distribution compound and eventually push the policy away
from the dataset support (Kumar et al., 2020; Fujimoto &
Gu, 2021). The negative impact on performance and safety
is even more pronounced in physical robotic systems (Ada
et al., 2024; Zheng et al., 2024; Mao et al., 2024), which
necessitates curated low-level safeguards (Chi et al., 2023).

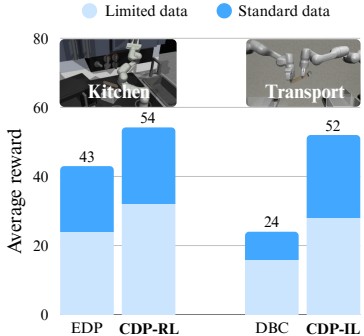

Figure 1: CDP's performance com-
pared to diffusion policy baselines in
offline RL and IL tasks.

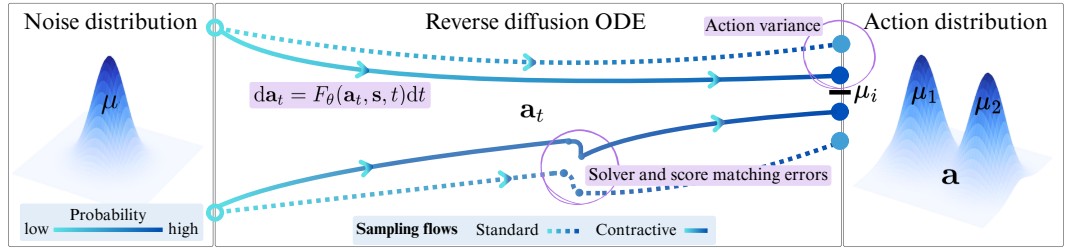

Figure 2: **Contraction in diffusion sampling.** We observe that contraction pulls nearby diffusion flows closer. Contraction plays a critical role in diffusion sampling for offline learning: it dampens solver and score-matching errors while reducing unwanted variance in the generated actions.

While the score-based and iterative sampling is crucial for capturing diverse action distributions, it introduces inaccuracies that must be addressed. To this end, we introduce **C**ontractive **D**iffusion **P**olicies (CDPs) to learn conditional action distributions and encourage *contraction* in the diffusion sampling process. Contraction theory studies whether the solutions of a differential equation converge over time, enabling the system to quickly forget small perturbations in its initial conditions and naturally suppress error growth (Tsukamoto et al., 2021; Dawson et al., 2023). Theoretically, contraction is proven to reduce solver and score-matching imprecisions in diffusion modeling under step-size bounds (Tang & Zhao, 2024). As shown in Figure 2, CDPs leverage contraction to tackle solver and score-matching errors in offline policy learning while reducing unwanted action variance by pulling the nearby solver flows closer to the dominant modes of the action distribution.

Built on our rigorous theoretical analysis, we derive a practical recipe by adding only *one tuned hyperparameter* and a *computationally efficient contraction loss* to promote contraction in the sampling process of diffusion policies. As a result, CDPs are straightforward to implement and seamlessly integrate with existing architectures. We empirically validate CDPs through detailed experiments on D4RL (Fu et al., 2020) and Robomimic (Mandlekar et al., 2021) benchmarks, as well as on physical robotic manipulation tasks. As highlighted in Figure 1, CDPs often outperform non-contractive diffusion policies, particularly in low-data regimes. Such results are consistent with the hypothesis that contraction mitigates the impact of score-matching and solver errors, ultimately enhancing the performance of standard diffusion policies.

**Summary of related work.** Recent studies have extensively applied diffusion policies to offline learning. In offline RL, DQL (Wang et al., 2023), EDP (Kang et al., 2023), and IDQL (Hansen-Estruch et al., 2023) combine behavior regularization through diffusion loss with value maximization, while hierarchical approaches (Ma et al., 2024) and divergence minimization methods (Gao et al., 2025) have been proposed to enhance scalability and stability. In IL, DP (Chi et al., 2023) and DBC (Pearce et al., 2023) model conditional action distributions, and 3D-aware variants integrate visual, language, and proprioceptive modalities (Ze et al., 2024; Ke et al., 2024). The emphasis in these works has largely been on training pipelines and multimodal integration rather than the diffusion process dynamics. Contraction-based perspectives remain relatively underexplored, with methods like contractive autoencoders (Rifai et al., 2011) demonstrating reduced sensitivity to input perturbations in representation learning. More recent contractive diffusion probabilistic models (Tang & Zhao, 2024) show that enforcing contraction can mitigate score-matching and discretization errors. While insightful, the method enforces contraction globally, potentially reducing diversity and hindering efficient integration in offline learning. A more in-depth literature review is presented in Appendix A.

## 2 BACKGROUND: OFFLINE LEARNING, DIFFUSION, AND CONTRACTION

Consider an environment modeled as a Markov Decision Process (MDP). The environment dynamics are governed by an *unknown* transition distribution $\mathbb{P}(\mathbf{s}' \mid \mathbf{s}, \mathbf{a})$, where taking an action $\mathbf{a} \in \mathcal{A} \subseteq \mathbb{R}^{d_\mathbf{a}}$ while in state $\mathbf{s} \in \mathcal{S} \subseteq \mathbb{R}^{d_\mathbf{s}}$ results in a transition to the next state $\mathbf{s}' \in \mathcal{S}$. The transition yields a reward signal, given by $R(\mathbf{s}, \mathbf{a})$. A policy $\pi : \mathcal{S} \times \mathcal{A} \to [0, 1]$ assigns a probability distribution over actions given a state. When parameterized by $\theta$, $\pi_\theta(\mathbf{a} \mid \mathbf{s})$ represents the probability of selecting action $\mathbf{a}$ given state $\mathbf{s}$ under the parameterized policy. We assume the existence of a dataset, $\mathcal{D} = \{(\mathbf{s}, \mathbf{a}, \mathbf{s}', r)\}$, collected from the interactions of a behavior policy $\pi_b$ with observed rewards $r$.

IL methods often consider the behavior policy $\pi_b$ as an expert and aim to learn $\pi_\theta$ to imitate the expert's behavior (Hussein et al., 2017). On the other hand, offline RL makes no such assumption, $\pi_b$ may be unknown or suboptimal. The goal is to learn $\pi_\theta$ from the offline dataset $\mathcal{D}$ that maximizes the expected discounted return for the environment (Levine et al., 2020).

## 2.1 DIFFUSION MODELING USING DIFFERENTIAL EQUATIONS

Diffusion policies have been employed to learn the true underlying policy distribution $\pi_\theta \approx p(\mathbf{a} \mid \mathbf{s})$ from offline data $\mathcal{D}$ (Chi et al., 2023; Dong et al., 2025). The *forward* diffusion process is first designed to gradually corrupt the dataset action $\mathbf{a} \equiv \mathbf{a}_0$ into a noisy version $\mathbf{a}_t$, where the diffusion step $t \in [0, 1]$. The process is modeled by an SDE:

$$d\mathbf{a}_t = f(\mathbf{a}_t, t)dt + g(t)d\mathbf{w}_t, \quad \mathbf{a}_0 \sim p(\mathbf{a}), \tag{1}$$

where $f : \mathbb{R}^{d_\mathbf{a}+1} \to \mathbb{R}$ is the drift function, $g : \mathbb{R} \to \mathbb{R}$ is the diffusion coefficient which scales the noise, $\mathbf{w}_t$ is the increment of a Wiener process (Song et al., 2021b), and $p(\mathbf{a}) = \int p(\mathbf{a} \mid \mathbf{s})p(\mathbf{s}) \, d\mathbf{s}$ is the true action marginal distribution. The forward SDE seen in Equation 1 has an equivalent *reverse sampling process* that maps the noisy input $\mathbf{a}_1$ back to the original sample $\mathbf{a}_0 \equiv \mathbf{a}$, described by:

$$d\mathbf{a}_t = \left[ f(\mathbf{a}_t, t) - g(t)^2 \nabla_{\mathbf{a}_t} \log p_t(\mathbf{a}_t \mid \mathbf{s}) \right] dt + g(t) \, d\bar{\mathbf{w}}_t, \tag{2}$$

where $d\bar{\mathbf{w}}_t$ is a Wiener process in reverse time. Note that the score function is the guiding hand which steers the denoising flows towards high-density regions of the true distribution based on the gradient of the log-likelihood of actions given a state at each step $t$. We focus on an equivalent *ordinary differential equation (ODE)* diffusion sampling for efficient sampling.

**Theorem 2.1.** *(Song et al., 2021b) There exists an equivalent probability flow ODE with the same marginal distribution, but better sample efficiency as the SDE in Equation 2, given by:*

$$d\mathbf{a}_t = \left[ f(\mathbf{a}_t, t) - \frac{1}{2} g(t)^2 \nabla_{\mathbf{a}_t} \log p_t(\mathbf{a}_t \mid \mathbf{s}) \right] dt. \tag{3}$$

Note that the only unknown component here is the score function, as all other terms in the reverse process are determined by the forward diffusion parameters (Dong et al., 2024a). Equation 3 generates reverse flows $\mathbf{a}_1 \to \ldots \mathbf{a}_t \ldots \to \mathbf{a}_0 \equiv \mathbf{a}$ which, given an accurate score, approximates the true policy distribution. The objective is then to learn a scaled score function, by optimizing

$$\mathcal{L}_\mathrm{d}(\theta) := \mathbb{E}_{t \sim \mathcal{U}(0,1), \, \mathbf{a}_t \sim p_t(\cdot \mid \mathbf{s}_t)} \, \| \epsilon_\theta(\mathbf{a}_t, \mathbf{s}, t) + \sigma_t \nabla_{\mathbf{a}_t} \log p_t(\mathbf{a}_t \mid \mathbf{s}) \|_2^2, \tag{4}$$

where $\sigma_t$ is the known standard deviation of the noise at a time $t$ in the forward process. In practice, we only have samples of $p_t(\mathbf{a}_t \mid \mathbf{s})$, and expect to train a highly accurate scaled score, $\epsilon_\theta(\mathbf{a}_t, \mathbf{s}, t) \approx -\sigma_t \nabla_{\mathbf{a}_t} \log p_t(\mathbf{a}_t \mid \mathbf{s})$ to approximate the true action distribution.

**Limitations of score-based learning**. In practice, limitations in sample count and diversity directly impact the quality of the learned score function. Additionally, the sampling process described in Equation 3 typically relies on SDE/ODE solvers (Zheng et al., 2023) or discrete samplers (Song et al., 2021a) for generating the iterative diffusion flows. These solvers are prone to discretization and integration errors due to finite step sizes, with a potential accumulation of approximation errors over denoising iterations.

**Offline learning with diffusion policies.** Offline learning methods adopt diffusion models to learn $\pi_\theta$ in which the actions are generated through a state-conditioned reverse diffusion process. Offline RL methods (Wang et al., 2023; Kang et al., 2023) enhance the score-matching loss in Equation 4 by incorporating value maximization, encouraging actions that balance proximity to $\pi_b$ and high expected returns. Diffusion policies (Chi et al., 2023; Ke et al., 2024), meanwhile, directly model $\pi_\theta$ through maximum likelihood training on state-action pairs. Since both paradigms build on diffusion models, they are reliant on accurate score functions and efficient ODE solvers for best performance.

## 2.2 CONTRACTION THEORY IN DIFFERENTIAL EQUATIONS

Consider an ODE in continuous time $d\mathbf{a}_t = F(\mathbf{a}_t, t)dt$. Contraction theory characterizes the convergence behavior of ODE flows $\mathbf{a}_t$ as $t$ increases (Lohmiller & Slotine, 1998; Tsukamoto et al., 2021). By regulating the rate of contraction, trajectories of a contractive ODE become robust to perturbations and compounding errors. Contraction is formally defined as follows.

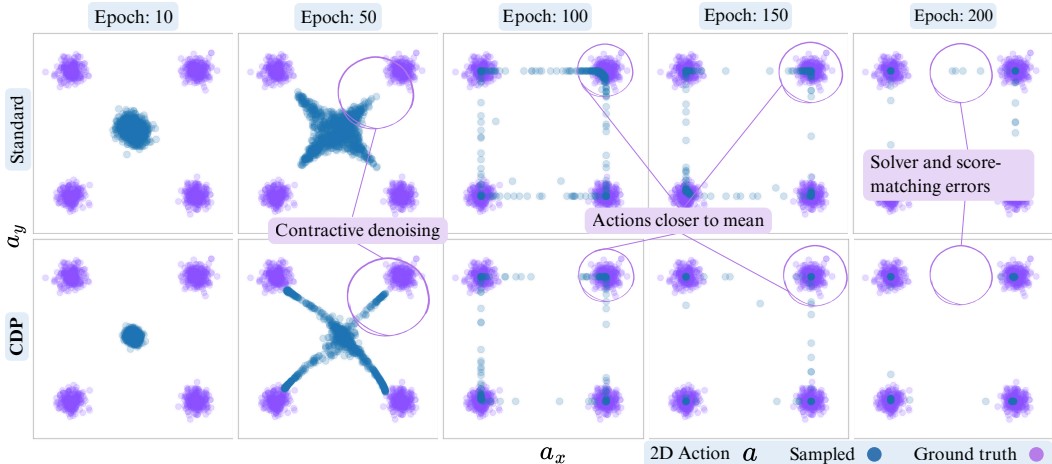

Figure 3: **2D toy experiments.** As we train CDP and a standard policy, both methods produce increasingly accurate actions. However, the actions generated by CDP tend to concentrate near the mean of distinct action modes, and show signs of mitigating solver and score-matching drifts.

**Definition 2.1.** *(Lohmiller & Slotine, 1998) An ODE, $F(\mathbf{a}_t, t)$, is contracting with a contraction rate $\eta \in \mathbb{R}_+$, if for any two initial conditions, $\mathbf{a}_0^1, \mathbf{a}_0^2 \in \mathbb{R}^{d_\mathbf{a}}$, and some constant $c \in \mathbb{R}_+$, the $\{\mathbf{a}_t\}_{t \in [0, +\infty]}$ flows generated by $F$ for each initial condition satisfy*

$$\left\| \mathbf{a}_t^1 - \mathbf{a}_t^2 \right\| \leq c\, e^{-\eta t} \left\| \mathbf{a}_0^1 - \mathbf{a}_0^2 \right\| \quad \forall t \geq 0,$$

*where $\| \cdot \|$ denotes an $L_p$ norm.*

Contraction can be analyzed by examining the Jacobian $J_F = \frac{\partial}{\partial \mathbf{a}_t} F(\mathbf{a}_t, t)$, which captures how the distance between nearby ODE flows evolves (Tsukamoto et al., 2021). As shown in Appendix D.3, a sufficient condition for contraction is

$$\lambda_{\max}\left(J_F + J_F^\top\right) < 0 \quad \forall t \geq 0, \tag{5}$$

where $\lambda_{\max}$ returns the largest eigenvalue of the symmetric $J_F$ for all $\mathbf{a}_t$ and $t$.

## 3 CONTRACTIVE DIFFUSION POLICIES

As discussed in Section 2.1, diffusion policies can generate state-conditioned actions by progressively steering noise toward actions through solving the reverse diffusion ODE. Our **key insight** is to *promote contraction while sampling from the reverse diffusion ODE* to mitigate score-matching and solver errors, as highlighted by the toy example in Figure 3. We pursue this goal in two steps: (i) providing a theoretical analysis of contraction in the diffusion ODE, and (ii) by introducing practical training techniques designed to actively promote contraction during learning.

### 3.1 CONTRACTION IN REVERSE DIFFUSION ODE

According to Section 2.2, we need to compute the Jacobian of the reverse diffusion ODE with respect to actions. Following the literature (Zhang & Chen, 2022), we reformulate the reverse diffusion ODE in Equation 3 to have a linear drift term $f(t)\mathbf{a}_t$ and view the approximate reverse process as

$$\mathrm{d}\mathbf{a}_t = [f(t)\mathbf{a}_t + h(t)\epsilon_\theta(\mathbf{a}_t, \mathbf{s}, t)]\,\mathrm{d}t = F_\theta(\mathbf{a}_t, t)\mathrm{d}t, \tag{6}$$

where $h(t) = g(t)^2 (2\sigma_t)^{-1}$ accounts for both the diffusion and noise schedule terms. Note that we consider Equation 6 with a fixed state without loss of generality, as $\mathbf{s}$ is only set once in the iterative reverse process to generate each action. In the next step, we calculate the Jacobian with respect to $\mathbf{a}_t$, which reveals the relationship between the score Jacobian and that of the reverse diffusion process:

$$J_{F_\theta} = \frac{\partial}{\partial \mathbf{a}_t} F_\theta(\mathbf{a}_t, t) = f(t)I + h(t)\frac{\partial \epsilon_\theta(\mathbf{a}_t, \mathbf{s}_t, t)}{\partial \mathbf{a}_t} = f(t)I + h(t)J_{\epsilon_\theta}. \tag{7}$$

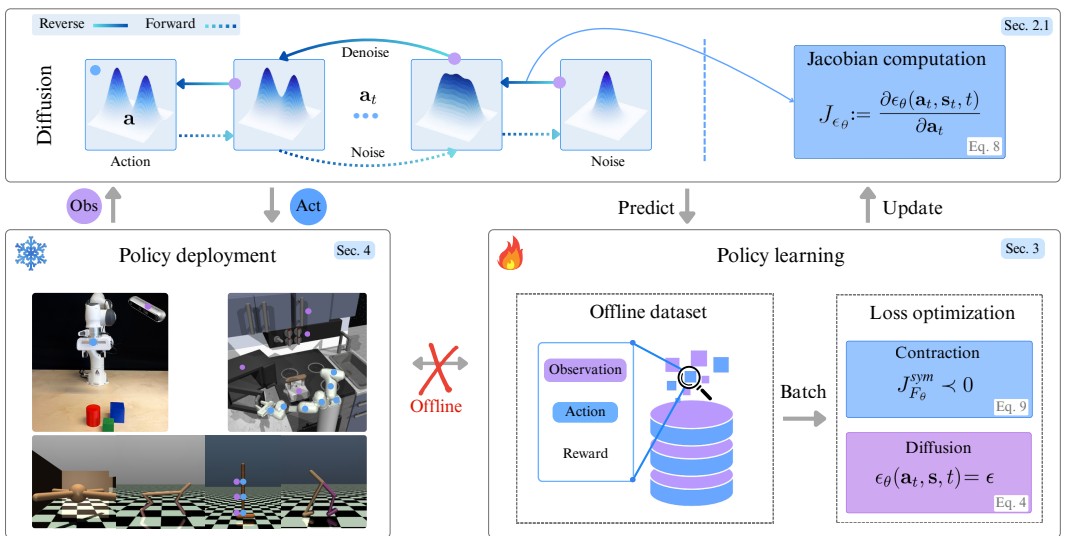

Figure 4: **Methodology overview.** The policy is trained on offline data to minimize contraction and diffusion losses. For each batch of data, the score Jacobian $J_{\epsilon_\theta}$, is efficiently computed for all denoising steps, and is then penalized with the contraction loss. At deployment, the diffusion policy is frozen, and the ODE sampling process generates the actions given observations.

The decomposition in Equation 7 is insightful. It shows that for any state, the contraction or expansion behavior of the diffusion ODE is governed by $f(t)I$ and $h(t)$. These are determined by scheduling parameters of the forward process, and $J_{\epsilon_\theta}$, which modulates the reverse diffusion flows based on the learned structure in the data. Notably, *only $J_{\epsilon_\theta}$ is trainable* among these terms, signaling that proper constraints on $J_{\epsilon_\theta}$ during training can alter the contractive behavior of the sampling process.

Noting the sufficient condition for contraction in Equation 5, we can examine the symmetric part (denoted by sym) of the matrix $J_{F_\theta}$, through $J_{F_\theta}^{\text{sym}} = \frac{1}{2}(J_{F_\theta} + J_{F_\theta}^\top) = f(t)I + h(t)J_{\epsilon_\theta}^{\text{sym}}$. The next theorem solidifies the results by highlighting the connection between the eigenvalues of $J_{\epsilon_\theta}^{\text{sym}}$ to the contraction of the sampling ODE.

**Theorem 3.1** (Interplay of score Jacobian and contractive sampling)**.** *Given a state $\mathbf{s} \in \mathcal{S}$ and a diffusion ODE, $\mathrm{d}\mathbf{a}_t = F_\theta(\mathbf{a}_t, \mathbf{s}, t)\mathrm{d}t$, $F_\theta$ is contractive, i.e., $J_{F_\theta}^{\text{sym}} \prec 0$, iff the **score Jacobian** satisfies*

$$\lambda_{\max}(J_{\epsilon_\theta}^{sym}) < -f(t)h(t)^{-1}, \quad \forall t \in [0, 1], \tag{8}$$

*where $\lambda_{\max}$ denotes the largest eigenvalue, and $f, h$ are defined by the forward process.*

The proof is outlined in Appendix D.2. Theorem 3.1 shows that the drift term $f(t)I$ is often contractive and pulls nearby flows closer together at each solver iteration. In contrast, the score term can locally exhibit either expansive or contractive behavior. Theorem 3.1 ensures that any local expansiveness of the score term is dominated by the contractive effect of the drift.

**Corollary 3.1.1** (Bounded action variance under contraction)**.** *For any two flows $\mathbf{a}_t^1$ and $\mathbf{a}_t^2$ initialized at $\mathbf{a}_0^1$ and $\mathbf{a}_0^2$, the difference in actions generated by $F_\theta$, denoted by $\delta\mathbf{a}_t = \mathbf{a}_t^1 - \mathbf{a}_t^2$, is bounded by*

$$\|\delta\mathbf{a}_t\| \;\leq\; \exp\!\Big(\int_t^1 \lambda_{\max}(J_{F_\theta}^{sym}(\tau))\,\mathrm{d}\tau\Big)\,\|\delta\mathbf{a}_0\|,$$

*where $J_{F_\theta}^{sym}(\tau)$ is the sampling ODE's Jacobian at solver step $\tau$.*

Appendix D.3 contains the proof. Corollary 3.1.1 describes a natural by-product of Theorem 3.1 as an upper bound for sensitivity to initial seed. Bounded sensitivity results in more robust action generation given similar states.

## 3.2 LEARNING RECIPE FOR CDPs

In Theorem 3.1, we have established that the contractive behavior of the reverse ODE process is linked to the largest eigenvalue of the negative score Jacobian, $\lambda_{\max}(J_{\epsilon_\theta}^{\text{sym}})$. Yet, direct penalization of

$\lambda_{\max}$ is computationally expensive, and critically, *excessive penalization could fuel a mode collapse in sampled action*. As illustrated in Figure 4, we design a loss to encourage contractive behavior in diffusion ODE while avoiding direct eigenvalue calculation and over-penalization. As a result, contraction is promoted by augmenting a penalty on $\lambda_{\max}(J_{\epsilon_\theta}^{\mathrm{sym}})$ into the diffusion loss to strike a perfect balance between denoising accuracy and encouraging contraction.

**(I) Efficient eigenvalue computation.** The first step is to compute the largest eigenvalue of $J_{\epsilon_\theta}^{\mathrm{sym}}$. We avoid direct and expensive computation of *all* eigenvalues using the following approximation.

**Lemma 3.1** (Power iteration (Austin et al., 2024)). *Let* $(J_{\epsilon_\theta}^{sym}) \in \mathbb{R}^{d \times d}$ *be symmetric with* $\lambda_{\max}$ *as the largest eigenvalue. For an initial vector* $v_0 \in \mathbb{R}^{d \times 1} \sim \mathcal{N}(0, I)$:

$$v_{k+1} \;=\; \frac{(J_{\epsilon_\theta}^{sym}) \, v_k}{\|J_{\epsilon_\theta}^{sym} v_k\|_2}, \qquad \hat{\lambda}_{k+1} \;=\; v_{k+1}^\top (J_{\epsilon_\theta}^{sym}) \, v_{k+1}, \quad k = 0, \dots, K-1.$$

*Then, for every* $k \geq 0$, $\left|\hat{\lambda}_k - \lambda_{\max}\right| \to 0$ *at a linear rate.*

Each backpropagation-friendly iteration of Lemma 3.1 costs only a single Jacobian-vector product. In practice, Appendix C.5 establishes that $K = 3$ or $4$ suffices for a stable penalty term, while significantly reducing computational and memory overheads.

**(II) Contraction loss.** With computational efficiency ensured, the next step is to formulate the contraction loss. Let $\hat{\lambda}_{\max}$ be the power-iteration estimate of $\lambda_{\max}(J_{\epsilon_\theta}^{\mathrm{sym}})$, and let $\hat{\lambda}_{\max}(J_{\epsilon_\theta}^{\mathrm{sym}})$ denote its value for a specific denoised action and step. Therefore, the per-sample contraction loss becomes

$$\mathcal{L}_{\mathrm{c}}(\theta) := \max(-\beta, \, \hat{\lambda}_{\max}(J_{\epsilon_\theta}^{\mathrm{sym}}) + f(t)h(t)^{-1}),$$

where $\beta > 0$ is a desired margin to enforce stricter contraction. An efficient alternative loss for penalizing the largest eigenvalue is the Frobenius norm, formulated as

$$\mathcal{L}_{\mathrm{c}}(\theta) := \|J_{\epsilon_\theta}^{\mathrm{sym}} + \beta \, I\|_{Frob} = \sqrt{\mathrm{trace}\left((J_{\epsilon_\theta}^{\mathrm{sym}} + \beta \, I)^\top (J_{\epsilon_\theta}^{\mathrm{sym}} + \beta \, I)\right)}.$$

The link between the Frobenius norm and eigenvalues comes from the fact that the Frobenius norm is tied to a matrix's eigenvalues for symmetric matrices, as further explained in Appendix D.5. Therefore, penalizing large $\mathcal{L}_{\mathrm{c}}$ pushes the Jacobian towards a negative curvature, while a conservative $\beta$ prevents *extreme levels of contraction* by avoiding excessively negative eigenvalues.

**(III) Training loss.** We now build the training loss by adding the score-matching loss $\mathcal{L}_{\mathrm{d}}$, described in Equation 4, to the contraction loss $\mathcal{L}_{\mathrm{c}}$ as

$$\mathcal{L}(\theta) = \mathbb{E}_{(\mathbf{a},\mathbf{s}) \sim \mathcal{D}} \left[ \mathbb{E}_{t \sim \mathcal{U}(0,1), \, \mathbf{a}_t \sim p_t(\cdot|\mathbf{s})} \left[ \|\epsilon_\theta + \sigma_t \nabla_{\mathbf{a}_t} \log p_t(\mathbf{a}_t|\mathbf{s})\|_2^2 \, + \, \gamma \, \mathcal{L}_{\mathrm{c}}(\theta) \right] \right], \quad (9)$$

with a single positive hyperparameter $\gamma$ to determine the contraction strength. Note that the score Jacobian needs to be calculated at each solver step during the sampling process, and for each state-action pair in a batch of data. While a Jacobian penalty on the score function alone could drive the model toward a trivially contractive field, Equation 9 prevents it by penalizing the score to be accurate, thereby striking a balance between contraction and meaningful flows in the diffusion ODE. Hence, $\gamma \mathcal{L}_{\mathrm{c}}(\theta)$ is regularized to *enforce local contraction*, pulling nearby trajectories together to reduce variance and improve robustness, but it does not collapse distinct, well-separated modes.

**(IV) Offline learning integration.** Since the added contraction loss term is calculated from diffusion ODE, the implementation is independent of the choice of policy learning method and is straightforward as long as a differentiable score function is accessible. Here, our method builds on two computationally efficient learning approaches: EDP (Kang et al., 2023) for offline RL and DBC (Pearce et al., 2023) for IL. We leave the details to Appendix C.

## 4 EXPERIMENTS

Our theoretical results and prior findings in the literature (Tang & Zhao, 2024) emphasize that contractive sampling dampens solver and score-matching errors and reduces unwanted sampling variance, thereby improving the consistency of action diffusion. Building on this foundation, we aim to empirically address three key questions:

Table 1: **Offline RL experiments with D4RL data.** CDP is compared to diffusion-based offline RL methods trained on the D4RL benchmark. Average and standard deviation of normalized episode return for each environment, and overall average across all environments are reported (higher is better). The best performing results for each environment, determined by the mean, are highlighted.

| Dataset | Environment | BC | IQL | DQL | EDP | IDQL | CDP |
|---|---|---|---|---|---|---|---|
| Medium Expert (ME) | HalfCheetah | $49.1 \pm 2.5$ | $84.9 \pm 0.3$ | $91.2 \pm 0.2$ | $93.4 \pm 0.3$ | $88.9 \pm 0.1$ | $94.8 \pm 0.3$ |
| | Hopper | $45.9 \pm 3.1$ | $101.5 \pm 3.9$ | $106.0 \pm 8.2$ | $110.0 \pm 2.8$ | $107.4 \pm 2.0$ | $110.0 \pm 1.5$ |
| | Walker2D | $56.6 \pm 3.5$ | $109.4 \pm 2.0$ | $105.0 \pm 4.0$ | $109.8 \pm 0.4$ | $110 \pm 1.1$ | $109.4 \pm 0.6$ |
| Medium (M) | HalfCheetah | $44.5 \pm 4.3$ | $46.4 \pm 5.0$ | $49.1 \pm 2.1$ | $46.2 \pm 0.2$ | $48.3 \pm 2.6$ | $46.0 \pm 0.2$ |
| | Hopper | $53.4 \pm 4.8$ | $52.2 \pm 3.3$ | $36.6 \pm 7.4$ | $61.1 \pm 3.4$ | $54.2 \pm 1.7$ | $62.8 \pm 2.6$ |
| | Walker2D | $68.5 \pm 3.4$ | $76.2 \pm 2.5$ | $78.5 \pm 5.1$ | $81.7 \pm 0.2$ | $80.9 \pm 1.3$ | $86.5 \pm 0.5$ |
| Medium Replay (MR) | HalfCheetah | $24.6 \pm 2.4$ | $37.9 \pm 0.7$ | $47.8 \pm 0.6$ | $43.4 \pm 0.6$ | $42.4 \pm 0.3$ | $43.9 \pm 0.2$ |
| | Hopper | $18.9 \pm 5.4$ | $48.1 \pm 0.3$ | $50.4 \pm 4.7$ | $55.1 \pm 3.5$ | $51.5 \pm 0.1$ | $63.5 \pm 4.1$ |
| | Walker2D | $27.8 \pm 4.7$ | $73.8 \pm 4.9$ | $76.8 \pm 3.5$ | $82.0 \pm 2.6$ | $84.6 \pm 2.4$ | $89.7 \pm 0.8$ |
| Mixed Partial Complete | Franka Kitchen | $42.5 \pm 3.0$ | $59.9 \pm 8.0$ | $58.1 \pm 1.4$ | $57.2 \pm 1.8$ | $58.2 \pm 4.1$ | $61.0 \pm 1.0$ |
| | Franka Kitchen | $31.9 \pm 1.9$ | $47.2 \pm 5.1$ | $42.3 \pm 2.7$ | $44.5 \pm 1.8$ | $49.0 \pm 2.5$ | $48.7 \pm 1.9$ |
| | Franka Kitchen | $27.3 \pm 1.1$ | $22.6 \pm 7.4$ | $35.7 \pm 6.2$ | $32.9 \pm 1.5$ | $31.6 \pm 3.8$ | $51.0 \pm 1.7$ |
| Play Diverse | Antmaze Medium | $0.2 \pm 0.0$ | $17.5 \pm 11.3$ | $21.9 \pm 1.6$ | $14.3 \pm 7.1$ | $17.7 \pm 5.7$ | $20.4 \pm 6.7$ |
| | Antmaze Medium | $0.6 \pm 0.1$ | $17.7 \pm 9.9$ | $23.5 \pm 2.3$ | $25.6 \pm 10.9$ | $19.3 \pm 5.0$ | $31.8 \pm 8.5$ |
| **Average reward** | | $35.1 \pm 2.9$ | $54.9 \pm 4.5$ | $58.8 \pm 3.6$ | $61.2 \pm 2.6$ | $60.3 \pm 2.3$ | $65.7 \pm 2.2$ |
| **Time** (seconds / 100k steps) | | $3115 \pm 12$ | $3742 \pm 36$ | $12822 \pm 164$ | $4594 \pm 43$ | $6250 \pm 35$ | $5236 \pm 71$ |

**(Q1)** Can CDPs enhance offline learning by mitigating score-matching and solver errors?
**(Q2)** Are CDPs particularly effective in low-data regimes?
**(Q3)** How well do CDPs transfer to and perform in the physical world?

To answer **(Q1–Q3)**, we conduct offline learning experiments on standard and partial version of D4RL (Fu et al., 2020) and Robomimic (Mandlekar et al., 2021) datasets. The learned policies are evaluated in Gymnasium (Towers et al., 2023) or Robosuite (Zhu et al., 2020), depending on the task, and the results are compared against baseline methods.

**Highlights of the results.** We find that contractive sampling generally improves diffusion policies for offline learning. Beyond the contraction loss weight $\gamma$, results are not highly sensitive to other hyperparameters. While contractivity does not yield universal gains, and performance is modest in some environments, we observe substantial benefits in others. Given these positive trends, together with straightforward integration and reasonable computational overhead, we view CDPs as a practical and effective choice to enhance offline learning methods.

## 4.1 SETUP

**Datasets.** We select MuJoCo (Hopper, Walker2D, and HalfCheetah), Franka Kitchen (Complete, Partial, and Mixed), and Antmaze (Medium Play and Diverse) from D4RL datasets for offline RL. For the MuJoCo environments, we utilize three data subsets: expert, medium-expert, and medium, which represent different expertise of the behavioral policy. Additionally, we evaluate on Robomimic datasets featuring demonstrations from tabletop manipulation tasks: Lift, Can, Square, and Transport. We test both low-dimensional proprioceptive observations and high-dimensional image-based observations. Further details are provided in Appendix G.

**Baselines and backbones.** For offline RL, we compare CDP against DQL (Wang et al., 2023), EDP (Kang et al., 2023), and IDQL (Hansen-Estruch et al., 2023), while for IL benchmarks we consider Diffusion Policy (Chi et al., 2023) and DBC (Pearce et al., 2023) with different conditioning backbones. Across all settings, we use a diffusion SDE backbone with an ODE solver for the reverse process. To integrate state information, we adopt a residual MLP (Pearce et al., 2023), which is sufficient for the low-dimensional observation spaces of D4RL. For Robomimic, we use the same MLP encoder for low-dimensional states, while for image-based observations we obtain an embedding vector using diffusion transformers (Dong et al., 2024b). Appendix C provides further details about baseline implementation.

Table 2: **IL experiments with Robomimic data.** We compare CDPs against the baselines on different tasks with low-dimensional (L) and image-based (H) observation spaces. We report the average and variance of success rate. The overall average and two best performing results are highlighted.

| Dataset | Task | BC-GMM | DP-DiT | DP-Unet | DBC-DiT | CDP-DiT | CDP-Unet |
|---|---|---|---|---|---|---|---|
| Robomimic (Lowdim) | Lift-L | $0.83 \pm 0.03$ | $0.99 \pm 0.20$ | $1.00 \pm 0.10$ | $0.94 \pm 0.07$ | $1.00 \pm 0.05$ | $1.00 \pm 0.15$ |
| | Can-L | $0.67 \pm 0.09$ | $0.96 \pm 0.09$ | $0.98 \pm 0.13$ | $0.82 \pm 0.04$ | $0.97 \pm 0.05$ | $1.00 \pm 0.08$ |
| | Square-L | $0.44 \pm 0.04$ | $0.79 \pm 0.08$ | $0.80 \pm 0.08$ | $0.57 \pm 0.11$ | $0.56 \pm 0.10$ | $0.83 \pm 0.04$ |
| | Transport-L | $0.21 \pm 0.01$ | $0.56 \pm 0.04$ | $0.74 \pm 0.03$ | $0.13 \pm 0.01$ | $0.48 \pm 0.07$ | $0.81 \pm 0.06$ |
| Robomimic (Image) | Lift-H | $0.75 \pm 0.03$ | $0.99 \pm 0.19$ | $1.00 \pm 0.11$ | $0.91 \pm 0.18$ | $1.00 \pm 0.04$ | $1.00 \pm 0.07$ |
| | Can-H | $0.56 \pm 0.11$ | $0.93 \pm 0.11$ | $0.95 \pm 0.12$ | $0.84 \pm 0.11$ | $1.00 \pm 0.03$ | $1.00 \pm 0.12$ |
| | Square-H | $0.43 \pm 0.01$ | $0.46 \pm 0.01$ | $0.88 \pm 0.16$ | $0.62 \pm 0.06$ | $0.72 \pm 0.13$ | $0.82 \pm 0.09$ |
| | Transport-H | $0.14 \pm 0.03$ | $0.41 \pm 0.04$ | $0.68 \pm 0.03$ | $0.29 \pm 0.01$ | $0.52 \pm 0.05$ | $0.75 \pm 0.04$ |
| **Average** | | $0.50 \pm 0.04$ | $0.76 \pm 0.10$ | $0.88 \pm 0.10$ | $0.64 \pm 0.07$ | $0.78 \pm 0.07$ | $0.90 \pm 0.08$ |
| **Time** (seconds / 100k steps) | | $2635 \pm 14$ | $7880 \pm 43$ | $8210 \pm 51$ | $7141 \pm 48$ | $8554 \pm 67$ | $9056 \pm 53$ |

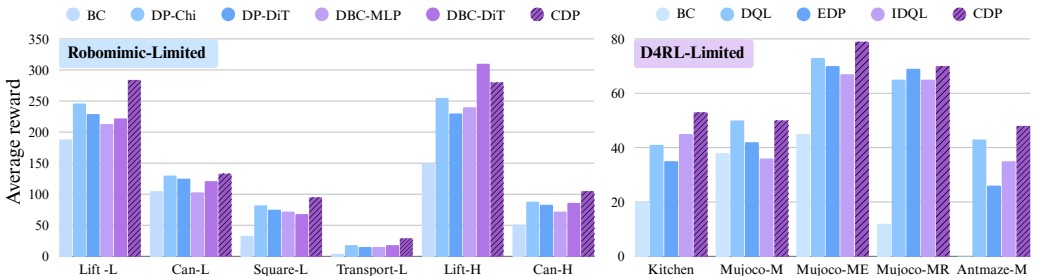

Figure 5: **Experiments on reduced datasets.** We report the average episode return across different random seeds. When training on only 10% of the original dataset size, we observe that CDP decisively outperforms the baselines. This improvement stems from the ability of contraction to dampen score-matching errors, which are amplified in low-data regimes.

**Training and evaluation.** Training takes 200k-500k gradient steps based on the task difficulty and observation modality. For each dataset, we train every method with 10 random seeds and report the average and standard deviation of either normalized episode returns or success rates. At train time, we monitor diffusion and contraction losses, along with policy and value loss for offline RL. We save trained policies at 20k intervals, and evaluate those policies in the same environment as the data is collected from. The hyperparameter $\gamma$ is naively tuned by picking the best performing $\gamma \in \{0.001, 0.01, \ldots, 100\}$, and we directly set $\gamma = 0.1$. Details are outlined in Appendix C.4.

## 4.2 KEY OBSERVATIONS

**General performance in offline learning (Q1).** We evaluate CDPs on offline RL benchmarks in Table 1. On average, CDP outperforms baselines across D4RL environments, with particularly strong gains in Kitchen and MuJoCo Medium-Replay datasets. In contrast, for tasks where all methods already achieve near-optimal rewards, CDP matches baseline performance, and in rare cases the additional robustness enforced by contractivity can slightly impede performance. In the IL setting, CDP consistently outperforms DBC, the method it builds upon, but trails DP-Unet. We attribute this gap to the architectural advantages of UNet-based models as well as their ability to generate longer action sequences, which is particularly beneficial in complex imitation tasks.

**Performance with limited data (Q2).** Reducing dataset size for each task introduces inaccuracies in score matching. Without contraction, these errors accumulate and propagate through the sampling process. By contrast, CDP maintains stability, which results in enhanced evaluation-time performance relative to baselines in Figure 5. This property is particularly valuable in practice, as real world data collection for offline learning is often costly and resource-intensive.

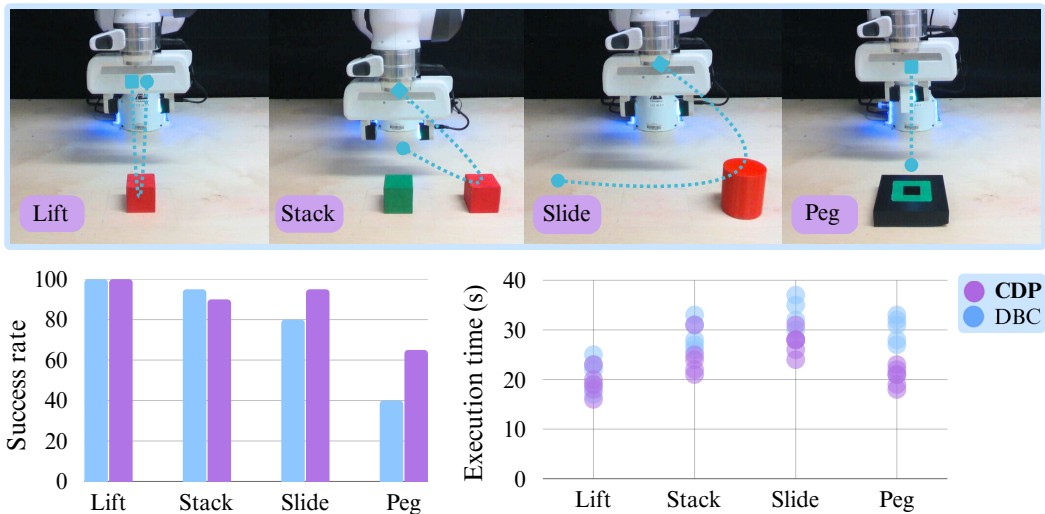

Figure 6: **Experiments with the physical Franka arm.** We execute each policy 20 times per task, and report the average success rate and execution time. CDP often performs better than standard DBC, particularly for the harder tasks: Slide and Peg.

**Physical robot experiments (Q3).** We evaluate CDP on four IL tasks: Slide, Stack, Peg, and Lift. Demonstration trajectories are collected via teleoperation, with observations consisting of agentview images and proprioceptive data, and actions represented as end-effector delta poses. As shown in Figure 6, CDP successfully solves three out of four tasks and achieves a higher success rate than DBC, including on the challenging Peg task. Further details on the experimental setup and data collection are provided in Appendix B and Appendix G.3, respectively.

**Supporting experiments.** Appendix F presents additional studies on the impact of different contraction loss weights, the behavior of CDP under intensified solver errors, and extended evaluation curves corresponding to the results in Table 1 and Table 2. Together, these results provide deeper insight into how contraction improves sampling quality in offline learning.

## 5 CONCLUSION

We probed the contractive behavior of the diffusion sampling process, leading to the development of CDPs, a class of diffusion policies with contractive sampling. We showed that contraction can be achieved by regularizing the score Jacobian, which can be efficiently integrated into the denoising diffusion backbone of offline learning methods. Our experiments demonstrated improved policy learning in standard continuous control tasks, as well as in scenarios with limited data.

**Limitations.** A key limitation of this study is the sensitivity of the diffusion ODE to the choice of contraction loss coefficient. While its value can be tuned by a light hyperparameter search, poorly tuned values can adversely impact the policy's performance. Moreover, CDP is built upon EDP or DBC based on the offline learning setting, and the effect of promoting contraction in other offline learning methods is not explored. Lastly, our work conducts limited experiments with image-based observations in simulation, and no low-dimensional experiments in real world setups.

**Future work.** Aside from addressing the limitations above, future studies could investigate the role of the score function's sensitivity to states given fixed actions. Additionally, we deliberately refrain from an extensive hyperparameter search for contraction-related parameters, but more systematic exploration could help identify theoretical ranges or bounds for these parameters, potentially linked to the structure of the data, observation and action spaces, or the choice of diffusion backbone. Finally, despite a brief study in Appendix E, we leave detailed studies of contraction in alternative diffusion sampling methods to the future work.

## ACKNOWLEDGMENT

We sincerely thank Wei-Di Chang, Harley Witzler, Mahrokh Ghoddousi Boroujeni, Hanna Yurchyk, and Stanley Wu for valuable feedback and discussions. This work was supported by the FRQNT Doctoral Training Scholarships and the NSERC Discovery Grant.

## REPRODUCIBILITY STATEMENT

In the spirit of full reproducibility, our codebase is attached and will be released publicly upon publication. In addition to the code, we provide the community with all configuration files and training scripts needed to regenerate every experiment, table, and figure in the paper. Each result is tied to the provided configuration, and we include dependency specifications and step-by-step run instructions to enable the straightforward execution of our pipelines or adapt them to new settings. With extensive documentation and a clean and straightforward codebase, our goal is to make verification frictionless such that the community can inspect, replicate, and build on our work with confidence. Lastly, in the appendix, we provide a step-by-step guide to implement the method, and present exact hyperparameters utilized to reproduce the reported performance.

## ETHICS STATEMENT

This work investigates contractive diffusion policies for offline learning with experiments conducted in a controlled lab. No human-subject data were collected beyond standard teleoperation by trained lab members; no personally identifying information was recorded or released. We consider potential misuse, for instance, deploying learned policies on unsafe hardware or in uncontrolled settings, and caution that our methods must not be used without a robotic experts to provide appropriate safeguarding and oversight of the deployment. Notably, we minimize environmental impact by using modest compute budgets and small hyperparameter sweeps, and we report implementation details to support reproducibility. There are no conflicts of interest or external sponsorships influencing the results of this work.

## CONTRIBUTIONS OF LLMS AND GENERATIVE AI

We used LLMs primarily for proofreading and improving the clarity and flow of the manuscript. In addition, LLMs assisted in extending our core theorem to alternative diffusion solvers (Appendix E) and in verifying as well as refining the presentation of proofs (Appendix D). On the implementation side, certain utilities and generic methods were generated with LLM support, while other classes and functions were linted and partially documented by similar agents. In literature review, LLMs were used to extensively search for related work and projects. *All LLM outputs were carefully cross-checked against credible references*, and no key theoretical developments or experimental results were produced by LLMs.

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

# Contractive Diffusion Policies: Appendix

## Table of Contents

# A    RELATED WORK

We present a more in-depth analysis of the related work. As our work is at the intersection of diffusion policies for offline learning, safety and robustness in action diffusion, and policy learning with dynamical systems, we divide this section accordingly.

## A.1    ACTION DIFFUSION FOR POLICY LEARNING

Diffusion modeling has been extensively used in policy learning to capture complex and multimodal action distributions. In *policy representation for offline RL*, DQL (Wang et al., 2023) is among the first to aim for training diffusion policies with a combination of behavior cloning and value maximization losses, boosting D4RL performance but at the cost of slow iterative denoising. DQL was immediately followed by EDP (Kang et al., 2023) which approximates actions from corrupted ones during training, and IDQL (Hansen-Estruch et al., 2023) which bridges diffusion modeling and implicit Q-learning for better trade-off between behavior regularization and value maximization. Others propose hierarchical (Ma et al., 2024) and multi-agent (Zhu et al., 2024) policies, as well as divergence minimization behavior regularization (Gao et al., 2025) to further tackle challenging offline RL scenarios.

Diffusion models have also been used for *planning in offline RL*. For instance, Diffuser (Janner et al., 2022) and AdaptDiffuser (Liang et al., 2023) are diffusion-based trajectory planners, with the former denoising offline trajectories into goal/constraint-guided plans and the latter adding an adaptive self-evolving mechanism to remain effective in changing environments. Lastly, Decision Diffuser (Ajay et al., 2023) treats decision-making as conditional generative modeling by sampling trajectories conditioned on desired returns or goals.

In *policy learning through imitation*, the focus is on perception, long-horizon structure, and multi-task generality. Diffusion policies were initially leveraged in DP (Chi et al., 2023) and DBC (Pearce et al., 2023) methods for learning conditional action distributions. DP3 (Ze et al., 2024) and 3D Diffuser-Actor (Ke et al., 2024) lift visuomotor diffusion from 2D pixels to point-clouds, improving robustness and generalization in manipulation. 3D Diffuser-Actor extends this by fusing language, proprioception, and 3D scene tokens with a denoising transformer to produce rich 6-DoF action trajectories across many tasks.

To enhance the existing diffusion policy methods, Consistency Policy (Prasad et al., 2024) leverages distillation to create a one-step policy from a pre-trained diffusion model, while OneDP (Wang et al., 2025) trains a single-step policy directly via diffusion distillation without a multi-step teacher. Furthermore, LDP (Xie et al., 2025) uses compact latent space representation and inverse dynamics models to build more accurate diffusion policies, while DiffAIL (Wang et al., 2024) proposes adversarial training for distributional alignment with expert demonstration. Lastly, methods such as Nomad (Sridhar et al., 2024) and Diffusion-EDF (Ryu et al., 2024) explore more specific applications in robotic navigation and manipulation.

## A.2    DIFFUSION GUIDANCE FOR ROBUST AND SAFE POLICIES

Safety in diffusion policies is relatively less explored. Safe Diffuser (Xiao et al., 2025) steers denoising with control-barrier guidance such that generated trajectories satisfy safety specs. Effectively, Safe Diffuser restricts the reverse diffusion process to generate samples belonging to a safe set. CoBL-Diffusion (Mizuta & Leung, 2024) fuses control barrier and Lyapunov functions to yield collision-free, stable plans in dynamic scenes. In contrast to Safe Diffuser, CoBL-Diffusion uses reward signals corresponding to safe actions to guide the diffusion process. More recently, Constrained Diffusers (Zhang et al., 2025) formulate the constrained diffusion generating process as constrained sampling, and solve it with constrained optimization techniques. These methods make safety or constraints a property of the reverse process itself, providing safety and reliability guarantees at generation time. However, learning these constraints or safety standards is still a challenging problem, and enforcing them causes a considerable computational overhead in high-dimensional spaces.

A different perspective is focused on feasibility and in-distribution guidance for training diffusion policies with offline RL. FISOR (Zheng et al., 2024) learns a feasibility critic to filter unsafe trajectories before execution, while searching for feasible actions with high rewards. Diffusion-

DICE (Mao et al., 2024) builds on DICE-style methods, adding in-sample, value-aware guidance that pulls samples toward high-return, in-distribution behaviors. This is achieved using a DICE guidance term interpreted as "how much more (or less) often will the policy visit each state-action than the data did", hence, keeping the policy close to the dataset support. Trajectory-level safe diffusion (Römer et al., 2024) alternates denoising with per-step constraint projections to ensure safe trajectory rollouts at test time. Lastly, SRDP (Ada et al., 2024) takes a different approach to address distribution shift by adding state representation objectives to harden policies to unseen states. Compared to the first perspective on safe diffusion policies, this line of work offers less computational overhead, but also looser generalization guarantees.

### A.3 CONTRACTION THEORY IN POLICY LEARNING

Contraction theory has been utilized to learn stable dynamical systems through IL for planning (Ravichandar et al., 2017; Abyaneh et al., 2025). Contraction in this scenario offers a more flexible stability criteria than using Lyapunov functions or diffeomorphism (Zhang et al., 2022; Abyaneh et al., 2024). In representation learning, Contractive Autoencoders (CAEs) (Rifai et al., 2011) leverage contraction theory to learn representations that are less sensitive to input perturbations, and showcase promising results at the cost of augmenting a simple and differentiable loss term. These representations can later be utilized to learn improved and more robust policies. CAEs (Rifai et al., 2011) shrink the representation changes when you nudge the input. CAEs explicitly penalize the encoder Jacobian to achieve this purpose, whereas CDP penalizes the score model's local sensitivity. In both cases the goal is discouraging high local gain and brittle behavior. Notably, CAE contraction is local around data, while CDPs need uniform contraction across reverse ODE trajectories.

**Contractive Diffusion Probabilistic Models.** Contraction theory has also been applied to improve the numerical stability of diffusion models, most notably in CDPMs (Tang & Zhao, 2024), which focus extensively on diffusion SDEs rather than ODEs. The central idea is to enforce contraction in the backward sampling process, hence, provably reducing both score-matching and discretization errors. This makes CDPMs more robust to inaccuracies in score estimation and time discretization. The approach can be implemented through transformations of pretrained model weights. However, contraction is imposed globally through specific design choices and assumptions on the drift and diffusion terms. Such hard constraints may completely ignore the local geometry of the score function, effectively suppressing diversity among dominant modes and limiting representational capacity. Furthermore, the formulation is not designed for conditional action diffusion or diffusion ODEs, making its integration into action diffusion for offline learning nontrivial.

### A.4 LIPSCHITZ REGULARIZATION FOR GENERALIZATION

Regularizing the Jacobian, or Lipschitz regularization, has been employed outside the scope of diffusion modeling to improve generalization and performance. For instance, (Jia & Su, 2020) define an information-theoretic metric based on the observed Fisher information to characterize local minima, prove a generalization bound in terms of this metric, and then use it directly as a regularizer to bias training toward enhanced minima. (Karakida et al., 2023) study gradient-norm regularization and show that a particular finite-difference scheme—implemented via alternating ascent/descent steps—both reduces the computational cost of gradient regularization and strengthens a desirable implicit bias toward rich-regime solutions. Lastly, (Foret et al., 2021) introduce sharpness-aware minimization, an efficient min-max optimizer that explicitly minimizes both loss and local sharpness by approximately optimizing the worst-case loss in a neighborhood of the current parameters. Yet, these methods mostly aim to improve generalization in deep neural networks and have not been extended to diffusion models.

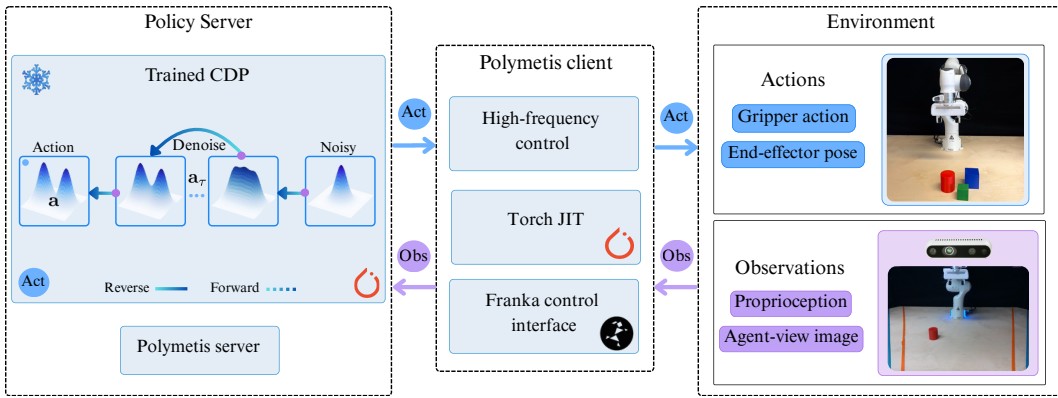

Figure 7: Physical robot setup for data collection and deployment of CDP. The trained and frozen policy denoises noisy action samples into control commands. Actions are communicated to the Polymetis middleware, which provides high-frequency control through Torch JIT and the Franka control interface. The Franka control interface executes gripper and end-effector actions while returning observations, including agentview and depth images plus proprioception, which are sent back to the policy for closed-loop continuous control.

# B REAL-WORLD EXPERIMENTS

## B.1 PHYSICAL SETUP

The experimental setup consists of a Franka Emika Panda[1] robot with 7 degrees-of-freedom and equipped with the Franka hand: a parallel two-finger gripper. The robot is mounted onto a workspace with dimensions of 1.8 m (length) × 1.22 m (width) × 0.82 m (height). As illustrated in the observation view of Figure 7, a RealSense D435i camera is positioned to capture the *agentview* perspective of the workspace. In this image, blue lines denote the camera field of view, while orange lines indicate the cropped region obtained after preprocessing, corresponding to the standard resolution of 224 × 224.

All models were trained on two RTX-4090s, each equipped with 24 GB of VRAM and allocated a maximum of 62.5 GB of CPU RAM. Despite this, a simple GPU with 4 GB of VRAM is sufficient to run a single experiment on low-dimensional observations, or to deploy an image-based diffusion policy with a ResNet backbone. As shown in Figure 8, the environments include Lift, Peg, Stack, and Slide, where Peg and Slide are more difficult due to precision requirements compared to the others.

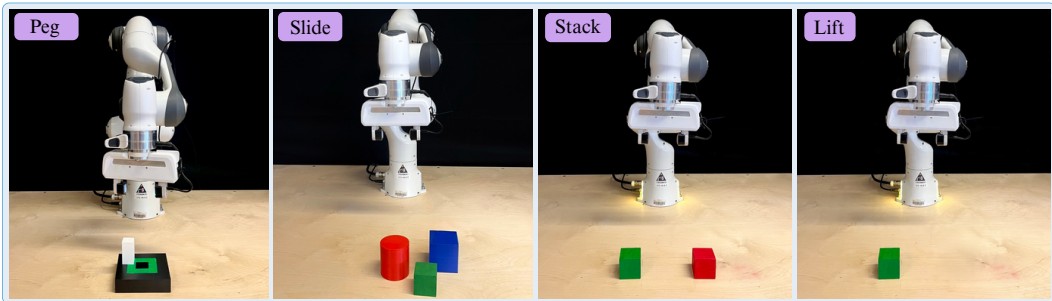

Figure 8: **Franka robot environments.** In comparison to other tasks, Peg and Slide are more difficult and require high precision, while Lift and Stack are relatively straightforward.

---

[1]https://franka.de/documents

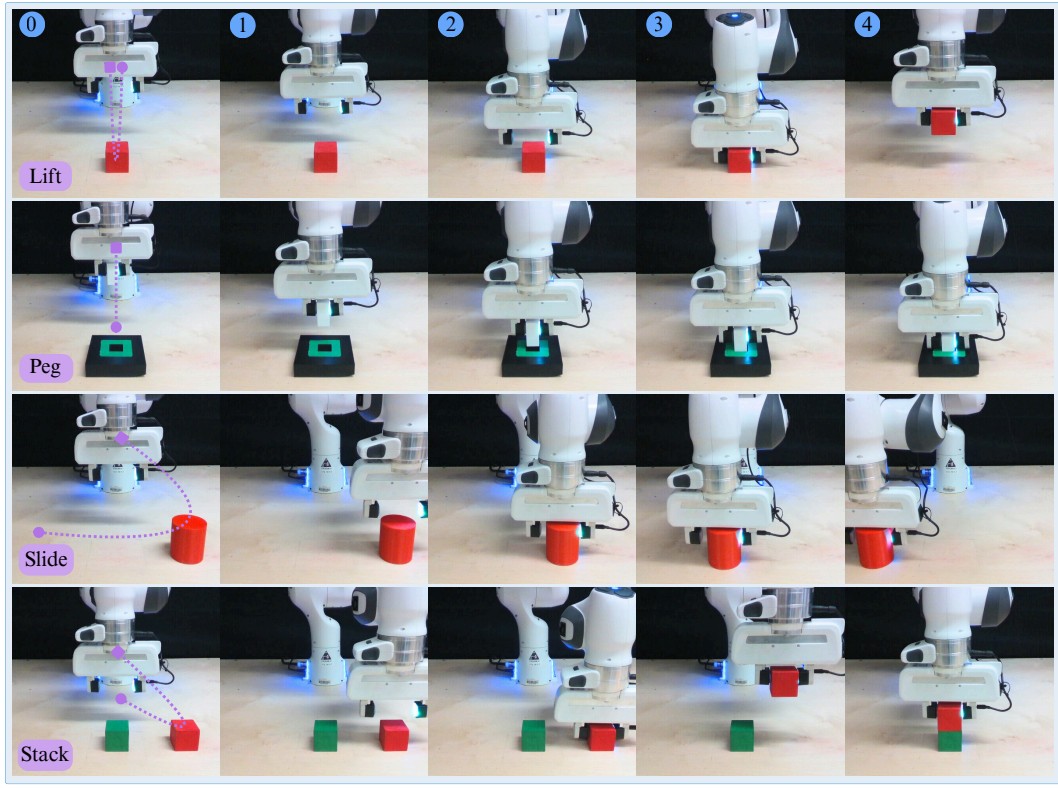

Figure 9: CDP is deployed for a set of 4 real-world tasks. The cropped footage is the real-time stream from the Intel RealSense camera, which communicates the image observations.

### B.2 DATA COLLECTION AND DEPLOYMENT

To evaluate our proposed method on real hardware, we collected teleoperated demonstrations on a Franka Emika Panda. CDP deployment on these tasks is shown in Figure 9. We use Polymetis [2] for high-level control and teleoperation; commands are executed on the Franka via `libfranka` over the Franka Control Interface (FCI) at 1 kHz, which returns current robot state to our controller. The end effector was teleoperated with a 3Dconnexion SpaceMouse [3], where 6-DoF inputs were mapped to Cartesian pose commands with adjustable gains. For each trajectory, we logged time-synchronized joint states (positions, velocities), end-effector pose, gripper width and state, RGB images, and the executed actions for each trajectory at a fixed 10 Hz rate. However, we only leverage *agentview* image and *proprioception* (end effector pose and gripper state) for training and deployment.

We deploy the trained policy using a two-machine client-server setup. On the robot's side, a client streams observations to a separate inference server that hosts the policy. The server runs inference and returns a 7-dimensional action vector (delta end effector position, orientation, and gripper state), which the client executes on the Franka via Polymetis.

---

[2] https://github.com/facebookresearch/fairo/tree/main/polymetis
[3] https://3dconnexion.com/br/spacemouse/

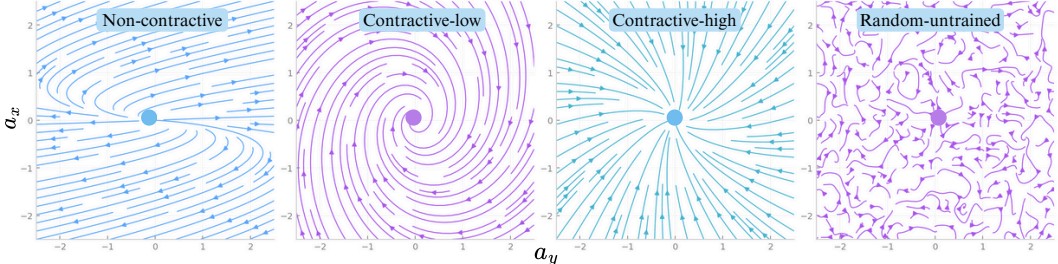

Figure 10: **ODE flows for a toy 2D action space.** The example is designed for a 2D action space with a fixed state, and portrays the way contraction affects the dynamics of diffusion ODE flows.

## C  IMPLEMENTATION DETAILS

In this appendix, we present a step-by-step recipe for promoting contraction in the diffusion ODE's sampling process, leveraging CDP's loss and efficient eigenvalue computation. Additional visualizations of the ODE flows in a toy 2D action space, comparing non-contractive and contractive variants, are provided in Figure 10. While our experiments focus on building CDPs using EDP (offline RL) and DBC (IL) algorithms, our contribution targets the sampling process, and hence, *it is compatible with any diffusion-based offline policy learning method*. We simply pick EDP and DBC for their efficient and straightforward implementation.

Note that CDP's codebase, as well as all baseline implementations, are built on top of `CleanDiffuser`. By providing a unified implementation of diffusion policy methods, the library enables fair and unbiased comparisons, ensuring that the only differences across methods are those explicitly specified in the configuration files, or the techniques used in each baseline, and not advantages such as conditioning backbones, solvers, sampling steps, and other factors related to diffusion modeling. To this end, we examine the `CleanDiffuser`[4] library in more detail, highlighting its comprehensive support for diffusion policy architectures and state-conditioning modules, and sharing insights into the behavior of different diffusion and conditioning backbones.

### C.1  BUILDING CDP FOR OFFLINE RL

**Method overview.** Efficient Diffusion Policy (EDP) (Kang et al., 2023) reduces the training-time overhead of diffusion-based policies such as DQL (Wang et al., 2023) by (i) using a *high-order sampler* to cut evaluation steps, and (ii) replacing multistep denoising inside the actor update with a *one-step* action approximation.

Let $\mathbf{a}_0$ denote the clean action and $\mathbf{a}_t$ its noised counterpart at diffusion time $t$ under the standard forward process (equivalent to Equation 1)

$$\mathbf{a}_t = \alpha_t \mathbf{a}_0 + \sigma_t \epsilon, \qquad \epsilon \sim \mathcal{N}(0, I). \tag{10}$$

During policy optimization, EDP avoids iterative reverse denoising of $\mathbf{a}_t$, as explained in Equation 3, and instead *approximates* $\mathbf{a}_0$ in one step:

$$\tilde{\mathbf{a}}_0 \approx x_\theta(\mathbf{a}_t, \mathbf{s}, t), \tag{11}$$

$$\tilde{\mathbf{a}}_0 \approx \frac{\mathbf{a}_t - \sigma_t \, \varepsilon_\theta(\mathbf{a}_t, \mathbf{s}, t)}{\alpha_t}, \tag{12}$$

where (11) is used with a data-prediction head and (12) follows directly from (10) when using a noise-prediction head (Ho et al., 2020). This replacement preserves useful gradients for the actor while eliminating the need to unroll many denoising steps inside the update. EDP also swaps the DDPM sampler for DPM-Solver (Zheng et al., 2023), a high-order ODE solver, enabling accurate sampling with a small number of steps which noticeably boosts the efficiency.

**Building CDPs on top of EDP for offline RL**

---

[4]https://github.com/CleanDiffuserTeam/CleanDiffuser

1. **Threshold.** We normally assume that the contraction condition is enforced at every solver step during the denoising process. However, the parameter `contr_steps` allows contraction to be applied only in the later stages of denoising, leaving the high-noise iterations unconstrained. This option is *not required*, but can be beneficial for handling more complex action distributions.

2. **Jacobian.** Form the symmetric Jacobian of the diffusion actor with respect to input actions as

$$J_{\epsilon_\theta} = \frac{\partial \epsilon_\theta(\mathbf{a}_t, \mathbf{s}, t)}{\partial \mathbf{a}_t}, \ J_{\epsilon_\theta}^{\mathrm{sym}} = \tfrac{1}{2}(J_{\epsilon_\theta} + J_{\epsilon_\theta}^\top).$$

This targets the part that governs the reverse ODE's contraction based on Theorem 3.1. $J_{\epsilon_\theta}$ can be computed via Equation 12, given the approximated action $\tilde{\mathbf{a}}_0$,

$$\frac{\partial \tilde{\mathbf{a}}_0}{\partial \mathbf{a}_t} \approx \frac{\partial}{\partial \mathbf{a}_t} \left( \frac{\mathbf{a}_t - \sigma_t \, \epsilon_\theta(\mathbf{a}_t, \mathbf{s}, t)}{\alpha_t} \right) = \frac{1}{\alpha_t}(I - \sigma_t J_{\epsilon_\theta}).$$

3. **Contraction.** Measure contractivity by penalizing the Frobenius norm or a stronger alternative by approximating the leading eigenvalue $\lambda_{\max}(J_{\epsilon_\theta}^{\mathrm{sym}})$ via power iteration as described in Lemma 3.1 and penalize it through Section 3.2. Note that the Jacobian needs to be computed for every sampling timestep, and for all state-action pairs in the current batch.

4. **Loss.** Combine the selected terms with policy and value loss terms for offline RL or the maximum likelihood loss for IL, similar to Equation 9, and tune the hyperparameter $\gamma$ with cross-validation.

## C.2 Building CDP for IL

**Method overview.** Diffusion Behavioral Cloning (DBC) (Pearce et al., 2023) frames imitation as conditional action diffusion: learn a denoiser that maps $(\mathbf{a}_t, \mathbf{s}, t)$ to $\mathbf{a}_0$, capturing the stochastic, multimodal nature of demonstrations. With the standard forward process, the model predicts either $\epsilon_\theta$ conditioned on the observation $\mathbf{s}$. DBC is agnostic to the perception stack: any visual encoder can feed $s$ (pixels or history) into the denoiser. In this work, we plug in finetuned ResNet backbones and use diffusion transformers. Another contribution of DBC is *Diffusion-X sampling* which biases selection toward intra-distribution, human-like actions: it keeps denoising for extra iterations.

**Recipe to build CDP on top of DBC.** The integration is straightforward, and we generally follow the same procedure as Appendix C.1 to compute the score Jacobian. Unlike EDP integration, however, approximate actions are not directly available for this purpose. Instead, we compute the Jacobian of the score function itself, which is relatively simple to implement with both MLPs and transformer-based architectures provided in `CleanDiffuser`.

## C.3 ContractiveDiffusionSDE Module Implementation

`ContractiveDiffusionSDE` module (implemented with the same name in our codebase) is at the heart of CDP's implementation. It is built upon the `ContinuousDiffusionSDE` class within `CleanDiffuser`, which represents a continuous-time variant of a variance preserving diffusion SDE model. It defines both training (adding noise to data and learning to reverse it) and inference (sampling new data points) procedures using continuous-time SDEs and ODEs depending on the solver's choice. We introduce this module to integrate contraction analysis in the diffusion SDE framework, as described in Appendix C.1 and Appendix C.2. Below, we provide a quick summary of major components and configurations of this implementation and how it preserves compatibility with various solvers, diffusion, and conditioning backbones.

**Diffusion and conditioning backbones.** The core components of our framework are the `nn_diffusion`, which serves as the primary backbone for predicting either the noise or the clean signal during the reverse process, and the `nn_condition`, which embeds the conditional observation inputs into a suitable representation space. Through `CleanDiffuser`, the `ContractiveDiffusionSDE` class flexibly supports a range of backbone architectures. For diffusion, we primarily experiment with MLPs, which provide lightweight function approximation for low-dimensional problems, and diffusion transformers, which capture long-range dependencies and complex temporal-spatial correlations critical in sequential decision-making. For conditioning, we adapt the architecture to the modality and dimensionality of the observations: identity mappings

Table 3: **CDP's hyperparameters, configurations, and their importance.** We list contraction-related parameters first. Among them, $\gamma$ is the most influential, as it directly impacts evaluation performance. The parameter `loss_type` also plays an important role, though its effect is primarily on computational efficiency rather than performance. The parameters listed under general are shared between CDP and the baselines used in our experiments.

| Parameter | Description | Importance | Range / choices |
|---|---|---|---|
| *Contraction-specific* | | | |
| $\gamma$ | Weight on contraction loss | **High** | $[10^{-3}, 100]$ (log-tune) |
| `loss_type` | Contraction penalty type | High | `jacobian` / `eigen` |
| $\beta$ (`contr_thr`) | Target eigenvalue margin | Medium | 0.1 |
| `contr_steps` | Steps with contraction penalty | Medium | 1.0 or 0.2×`sampling_steps` |
| `num_pi` | Power-iteration count | Medium | $[3, 5]$ |
| *General training & inference* | | | |
| `actor_lr`, `critic_lr` | Actor/Critic learning rates | High | $1 \times 10^{-4}$ (tune: $[3 \times 10^{-5}, 3 \times 10^{-4}]$) |
| `solver` | Reverse ODE solver (DPM) | Medium | `ode_dpmsolver++_2M` |
| `diffusion_steps` | Diffusion steps during training | Medium | 50 (typical: $[50, 100]$) |
| `sampling_steps` | Sampling steps during evaluation | Medium | 15 (typical: $[10, 30]$) |
| `ema_rate` | EMA decay | Medium | 0.999 (typical: $[0.995, 0.9999]$) |
| `nn_diffusion` | Diffusion backbone | Medium | `DiT` / `MLP` |
| `nn_condition` | Conditioning network | Medium | `ResNet` / `Identity` |
| `gamma` | RL discount factor | Low | 0.99 (typical: $[0.95, 0.995]$) |

for low-dimensional vector states, and convolutional ResNet-based encoders when working with high-dimensional image inputs. This modular design enables us to tailor the expressivity and inductive biases of the backbones to the structure of the task, while maintaining consistency with our experiments.

**Noise schedules.** Given an input sample, the implementation applies a forward diffusion step using the given noise schedule. In `CleanDiffuser`, two standard schedules are supported. The *linear* schedule increases the variance linearly with time, leading to a simple and stable forward process that has been widely adopted in early diffusion models. In contrast, the *cosine* schedule adjusts the variance according to a shifted cosine function, which slows down noise growth in the early steps and accelerates it later. We use the linear schedule in our experiments but our implementation of `ContractiveDiffusionSDE` supports both scheduling schemes.

**Solvers.** We employ ODE-based DPM solvers (Zheng et al., 2023), which directly integrate the reverse diffusion ODE rather than relying on stochastic updates, and are consistant with our theoretical framework. These solvers approximate the solution trajectory by applying high-order numerical integration schemes that balance accuracy with computational efficiency. In particular, we use `ode_dpmsolver++_2M`, a second-order multistep method that leverages two adjacent denoising steps for improved stability and precision.

## C.4 HYPERPARAMETERS & CONFIGURATIONS

All critical hyperparameters and configurations of our method are summarized in Table 3. While the table lists a broad set of parameters, it is important to emphasize that *only two of them are critical for contractive sampling*. Among these two, $\gamma$ is the decisive factor that substantially influences both training stability and final evaluation performance. The second contraction-related parameter, $\beta$, plays a more supportive role, serving primarily as a safeguard rather than a driver of performance. To further clarify their impact, in Table 4 we report the best-performing values for all parameters based on our empirical search.

**CDP's hyperparameters.** Our hyperparameter tuning procedure is intentionally kept simple to highlight the robustness of the approach. Specifically, for $\gamma$, we perform a search over the discrete set $\{0.001, 0.01, 0.1, 1.0, 10\}$, which is sufficient to capture its effect on performance across benchmarks. The value of $\beta$ and `loss_type` is fixed to 0.1 and `jacobian`, respectively, reflecting their relatively minor influence compared to the weight term. Other parameters are inherited from standard diffusion policy baselines without modification, ensuring that the *only meaningful difference* introduced by our method stems from the contraction regularization. Therefore, we ensure that the observed

Table 4: **Best performing contraction loss weights.** We summarize the results of our hyperparameter tuning for the contraction loss weight here. Note that the results correspond to experiments with standard datasets for each benchmark.

| Environment | Contraction loss weight ($\gamma$) |
|---|---|
| Antmaze (Medium-Play, Medium-Diverse) | (0.001, 0.001) |
| Franka-Kitchen (Partial, Mixed, Complete) | (100.0, 100.0, 10.0) |
| Mujoco-Medium (Half-Cheetah, Hopper, Walker) | (0.01, 1.0 , 10.0 ) |
| Mujoco-Medium-Expert (Half-Cheetah, Hopper, Walker) | (0.01, 0.001, 0.01) |
| Mujoco-Medium-Replay (Half-Cheetah, Hopper, Walker) | (0.01, ,0.1 0.001) |
| Robomimic-Lowdim (Lift, Can, Square, Transport) | (0.01, 0.1, 1.0, 0.1) |
| Robomimic-Image (Lift, Can) | (0.01, 0.001) |
| Real-world tasks (Lift, Stack, Slide, Peg) | (0.001, 0.001, 0.01, 0.01) |

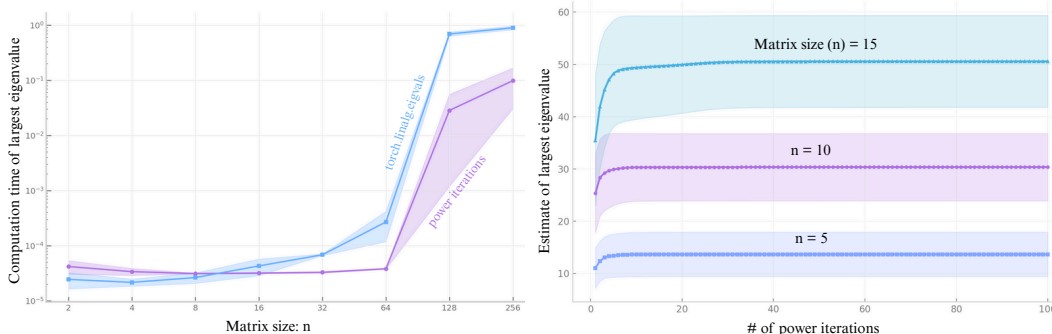

Figure 11: **Efficiency of power iterations for calculating the leading eigenvalue.** (Left) Computation time for $n \times n$ matrices compared against PyTorch's built-in eigenvalue method. (Right) Number of iterations required to obtain a stable estimate of the largest eigenvalue for different matrix sizes.

performance gains are not the product of extensive hyperparameter optimization, but rather a direct consequence of incorporating contraction into the learning process.

Note that the configuration files in the codebase specify the correct hyperparameters for each method. The only hyperparameter that requires tuning to achieve the best performance is the contraction loss weight, whose values are provided in Table 4.

## C.5 POWER ITERATIONS VS. DIRECT EIGENVALUE CALCULATION

The comparison between power iterations and PyTorch's direct eigenvalue computation highlights a clear difference in efficiency, particularly for large matrices. As shown in Figure 11, computing the largest eigenvalue via PyTorch's built-in routines becomes increasingly expensive as the matrix size grows, since it involves solving for the full spectrum of eigenvalues. Power iterations, in contrast, specifically target the leading eigenvalue and thus achieve markedly lower runtimes, particularly for medium to large matrices. This efficiency makes them appealing in scenarios where only the dominant eigenvalue is required, as is often the case in stability analysis or contraction-based regularization, where repeated evaluations must be carried out during training.

**Note.** For each batch of data, the number of times we need to calculate the largest eigenvalue is

$$\textbf{batch\_size} \times \textbf{num\_solver\_steps}(t),$$

which intensifies the differences observed in Figure 11.

We also show that only a small number of iterations are needed to reach a stable estimate of the largest eigenvalue, and the required iteration count remains modest even as matrix size increases. In practice, this means that one can obtain accurate eigenvalue estimates with just a handful of iterations, striking a favorable balance between computational cost and precision.

### C.6 BALANCING CONTRACTIVITY WITH MULTIMODAL LEARNING

Our policy is trained with the standard diffusion objective in Equation 4, which anchors the score to the data distribution and thus preserves task-relevant multimodality. The contraction loss integrated in Equation 9 **complements** this objective by tempering the sampler's seed/solver sensitivity in locally expansive directions of the reverse dynamics, improving reliability without collapsing distinct high-density modes. The delicate balance between contraction remains as long as over-regularization is avoided. In practice, our method provides more than sufficient tools to balance robustness and diversity with the following complementary knobs.

**Contraction (regularization) weight ($\gamma$).** As the key and only critical hyperparameter of CDP, $\gamma$ controls the overall strength of contraction and is tuned coarsely with a small grid (only 6 values in this work; see Section 4.1). The experiments showcase that for Robomimic and D4RL benchmarks, as well as our real-world tasks, these values are enough to showcase a reasonable performance gain, while all the other parameters are fixed. However, depending on the validation/test-time performance, one could opt for a more extensive search or simply leave the regularization term as a *bounded* trainable parameter to be learned with the diffusion loss.

**Contraction steps (`contr_steps`).** This parameter controls at which denoising steps the contraction regularizer is applied, allowing us to activate contraction only on a selected subset of (typically later) timesteps rather than throughout the entire sampling process. For instance, we can apply contraction only in the late denoising steps, which one could leverage when handling complex, highly multimodal action distributions. This provides a simple mechanism to **concentrate the trajectories near high-quality modes** in the final sampling stages, while **preserving expressivity and flexibility** of the diffusion model at earlier steps.

**Contraction threshold ($\beta$, `contr_thr`).** Finally, we note that the parameter $\beta$ controls the **maximum target contraction** level by acting as an eigenvalue threshold in our Frobenius-norm surrogate in Section 3.2. Concretely, we penalize violations of a bound of the form $\lambda_{\max}(J^{\mathrm{sym}}) \lesssim |J^{\mathrm{sym}}|_F \leq \beta$, so that smaller values of $\beta$ enforce stronger contraction (tighter eigenvalue bounds), while larger values allow more flexible dynamics. In practice, we set $\beta$ once to meaningfully reduce sensitivity without aggressively shrinking the action bandwidth or collapsing multimodality, and use only the contraction weight. Empirically, we observe that performance is reasonably stable with fixed $\beta$ values, which supports the robustness of our Frobenius-based eigenvalue thresholding in practice.

## D MATHEMATICAL PROOFS AND DERIVATIONS

### D.1 CONNECTING FORWARD DIFFUSION SDE TO SCHEDULING FUNCTIONS

The schedulers $\alpha_t$ and $\sigma_t$ originate from the non-SDE, but equivalent formulation of the forward diffusion process, where the noisy action at step $t$ is given by

$$\mathbf{a}_t = \alpha_t \mathbf{a}_0 + \sigma_t \epsilon, \quad \epsilon \sim \mathcal{N}(0, I). \tag{13}$$

These scheduling parameters control how the original and noise signals evolve in the forward process, and are therefore central to defining the trade-off between signal preservation and stochastic corruption during training. While they are inherited directly from standard diffusion policy formulations and remain fixed across our experiments, they provide the backdrop against which contraction-related modifications are introduced.

To establish the link between $\alpha$ and $\sigma$ and the forward SDE functions, $g$ and $f$, we start by recalling the forward diffusion SDE

$$\mathrm{d}\mathbf{a}_t = f(t)\,\mathbf{a}_t\,\mathrm{d}t + g(t)\,\mathrm{d}W_t,$$

its Itô solution is

$$\mathbf{a}_t = \alpha_t \mathbf{a}_0 + \int_0^t \frac{\alpha_t}{\alpha_s}\,g(s)\,\mathrm{d}W_s, \quad \alpha_t := \exp\left(\int_0^t f(u)\,\mathrm{d}u\right).$$

This implies

$$\mathbf{a}_t \,\big|\, \mathbf{a}_0 \sim \mathcal{N}\big(\alpha_t \mathbf{a}_0, \sigma_t^2 I\big),$$

with

$$\sigma_t^2 = \int_0^t \left(\frac{\alpha_t}{\alpha_s}\right)^2 g(s)^2\,\mathrm{d}s.$$

Conversely, given $\alpha_t > 0$ and $\sigma_t \geq 0$, the drift and diffusion are

$$f(t) = \frac{\mathrm{d}}{\mathrm{d}t}\log\alpha_t, \qquad g(t)^2 = \frac{\mathrm{d}}{\mathrm{d}t}\sigma_t^2 - 2\,f(t)\,\sigma_t^2.$$

This derivation connects the forward SDE process to scheduling functions, and enables us to find the drift and diffusion functions at a given diffusion step (Song et al., 2021a; Dong et al., 2024a).

### D.2 INTERPLAY OF SCORE JACOBIAN AND CONTRACTIVE SAMPLING ODE

**Theorem statement.** *Given a state $\mathbf{s} \in \mathcal{S}$ and a diffusion ODE, $\mathrm{d}\mathbf{a}_t = F_\theta(\mathbf{a}_t, \mathbf{s}, t)\mathrm{d}t$, $F_\theta$ is contractive, i.e., $J_{F_\theta}^{sym} \prec 0$, iff the score Jacobian satisfies*

$$\lambda_{\max}(J_{\epsilon_\theta}^{\mathrm{sym}}) < -f(t)h(t)^{-1}, \quad \forall t \in [0,1],$$

*where $\lambda_{\max}$ denotes the largest eigenvalue, and $f, h$ are defined by the forward process.*

*Proof.* According to Section 2.2, a deterministic flow $\mathrm{d}\mathbf{a}_t = F_\theta(\mathbf{a}_t, t)\mathrm{d}t$ is contractive with rate $\eta > 0$ in the L2-norm, if all eigenvalues of its symmetric Jacobian are negative, i.e.,

$$\lambda_{\max}(J_{F_\theta}^{\mathrm{sym}}) := \max_i \lambda_i\Big(\tfrac{1}{2}\big(J_{F_\theta} + J_{F_\theta}^\top\big)\Big) < 0 \quad \forall \mathbf{a}_t, t \in [0,1].$$

For the diffusion ODE $F_\theta$, we already know from the main text that $J_{F_\theta}^{\mathrm{sym}} = f(t)I + h(t)\,J_{\epsilon_\theta}^{\mathrm{sym}}$. Specifically for any two symmetric matrices A and B, by the special case of Weyl's inequality, the following holds for their maximum eigenvalues:

$$\lambda_{\max}(A + B) \leq \lambda_{\max}(A) + \lambda_{\max}(B).$$

Applying this to the Jacobian of the diffusion ODE, we get:

$$\lambda_{\max}(J_{F_\theta}^{\mathrm{sym}}) = \lambda_{\max}(f(t)I + h(t)\,J_{\epsilon_\theta}^{\mathrm{sym}}) \leq \lambda_{\max}(f(t)I) + \lambda_{\max}(h(t)\,J_{\epsilon_\theta}^{\mathrm{sym}}).$$

Assuming that the diffusion coefficient $h(t)$ is positive[5], sufficient condition for reverse ODE contraction boils down to

$$\lambda_{\max}(J_{F_\theta}^{\text{sym}}) \leq \underbrace{\lambda_{\max}(f(t)I)}_{\text{Drift}} + \underbrace{h(t)\,\lambda_{\max}(J_{\epsilon_\theta}^{\text{sym}})}_{\text{Diffusion}} < 0. \tag{14}$$

**Drift.** Note that since the drift term $f$ is uniformly dissipative, this is evident by

$$f(t) = \frac{\mathrm{d}\log\alpha_t}{\mathrm{d}t}, \quad \alpha_t \text{ is a dissipative schedule set in the forward process,}$$

$$\Rightarrow \quad \frac{\mathrm{d}\alpha_t}{\mathrm{d}t} < 0 \quad \Rightarrow \quad \frac{\mathrm{d}\log\alpha_t}{\mathrm{d}t} < 0 \quad \Rightarrow \quad f(t) < 0$$

$$\Rightarrow f(t)I \prec 0, \quad \text{since } \log \text{ is monotonically increasing and } f(t)I \text{ is diagonal.}$$

Hence, every $f(t)I$ is negative-definite, meaning that $\lambda_{\max}(f(t)I) < 0$. This is the usual situation for variance-preserving[6] (VP) schedules investigated in our work (Song et al., 2021a;b), where $\alpha_t$ is a dissipative function.

**Diffusion.** Revisiting Equation 14, we notice that with a positive $h$, the equation will be satisfied if

$$\lambda_{\max}(J_{\epsilon_\theta}^{\text{sym}}) < -\frac{\lambda_{\max}(f(t)I)}{h(t)} = \frac{-f(t)}{h(t)} \tag{15}$$

Since $f(t)I$ is a diagonal matrix with $f(t)$ as every entry on the diagonal, all of its eigenvalues are equal to $f(t)$. In simple terms, we just need an *upper bound* on $\lambda_{\max}(J_{\epsilon_\theta}^{sym})$. According to Appendix D.1, the bound in Equation 15 can be written in terms of signal and noise schedules as

$$\lambda_{\max}(J_{\epsilon_\theta}^{\text{sym}}) < \frac{-(2\sigma_t)\frac{\mathrm{d}}{\mathrm{d}t}\log\alpha_t}{(\frac{\mathrm{d}}{\mathrm{d}t}\sigma_t^2 - 2\left(\frac{\mathrm{d}}{\mathrm{d}t}\log\alpha_t\right)\sigma_t^2)} = \frac{-f(t)}{h(t)}. \tag{16}$$

Therefore, we can efficiently determine the bound $\frac{-f(t)}{h(t)}$ based on the known scheduling parameters of the forward diffusion process. $\qquad\square$

**Extension to variance-exploding (VE) schedules.** Our proof centers on VP-SDE schedules because they are the de facto choice in diffusion policies for control, they align with our experimental setups, and crucially, the probability flow ODE induced by VP has a uniformly dissipative drift. This negative drift cleanly maps into a stability margin in the symmetric Jacobian, yielding transparent contraction conditions and a direct diagnostic for seed/solver sensitivity.

For VE-SDEs, the diffusion ODE lacks the dissipative drift that VP provides, so stability no longer benefits from a built-in leftward spectral shift. However, it hinges entirely on curbing locally expansive directions of the learned score and the diffusion term in general. Hence, our contraction loss and the proof can still be applied: the only change required is to update the contraction threshold (using $\beta$ in Section 3.2) to adjust for the lack of dissipative drift term. Paired with a small $\gamma$, these controls balance robustness and multimodality under VE schedules, where the absence of dissipative drift could make the reverse flow more sensitive to score and integration errors.

### D.3 BOUNDED ACTION VARIANCE UNDER CONTRACTION

**Corollary statement.** *For any two flows* $\mathbf{a}_t^1$ *and* $\mathbf{a}_t^2$ *initialized at* $\mathbf{a}_0^1$ *and* $\mathbf{a}_0^2$, *the difference in actions generated by* $F_\theta$, *denoted by* $\delta\mathbf{a}_t = \mathbf{a}_t^1 - \mathbf{a}_t^2$, *is bounded by*

$$\|\delta\mathbf{a}_t\| \leq \exp\left(\int_t^1 \lambda_{\max}(J_{F_\theta}^{\text{sym}}(\tau))\,\mathrm{d}\tau\right)\|\delta\mathbf{a}_0\|,$$

*where* $J_{F_\theta}^{sym}(\tau)$ *is the sampling ODE's Jacobian at solver step* $\tau$.

---

[5]As defined in Equation 6, $h(t) = g(t)^2(2\sigma_t)^{-1}$. Hence, $h(t)$ is always positive since $\sigma$ represents a positive noise variance.

[6]For variance-exploding scheduling, the drift term loses any effect on the reverse process (Yang et al., 2023).

*Proof.* We start by following the steps to outline the sufficient condition for contraction on the largest eigen value (Lohmiller & Slotine, 1998): Given two trajectories $\mathbf{a}_t^1, \mathbf{a}_t^2$ of the system

$$\mathrm{d}\mathbf{a}_t = F(\mathbf{a}_t, t)\mathrm{d}t,$$

their infinitesimal separation at step $t \in [0, 1]$, defined by

$$\delta\mathbf{a}_t = \mathbf{a}_t^1 - \mathbf{a}_t^2$$

evolves as $\delta\dot{\mathbf{a}}_t = J(\mathbf{a}, t)\,\delta\mathbf{a}_t$, where $J(\mathbf{a}, t) := \frac{\partial F}{\partial \mathbf{a}}(\mathbf{a}, t)$. Tracking the squared distance in the *Euclidean metric*, $\delta\mathbf{a}^\top \delta\mathbf{a}$, yields

$$\frac{\mathrm{d}}{\mathrm{d}t}\left(\delta\mathbf{a}_t^\top \delta\mathbf{a}_t\right) = 2\delta\mathbf{a}_t^\top \delta\dot{\mathbf{a}}_t = 2\delta\mathbf{a}_t^\top J\delta\mathbf{a}_t.$$

Decompose $J$ into its symmetric part $J^{\mathrm{sym}}(\mathbf{a}_t, t) = \frac{1}{2}\left(J + J^\top\right)$. Let $\lambda_{\max}(\mathbf{a}_t, t)$ denote the largest eigenvalue of $J^{\mathrm{sym}}$. Given that for the asymmetric part of $J$, we have $\delta\mathbf{a}_t^\top J^{\mathrm{asym}}\delta\mathbf{a}_t = 0$ [7],

$$\delta\mathbf{a}_t^\top J\delta\mathbf{a}_t = \delta\mathbf{a}_t^\top J^{\mathrm{sym}}\delta\mathbf{a}_t \leq \lambda_{\max}(\mathbf{a}_t, t)\,\delta\mathbf{a}_t^\top \delta\mathbf{a}_t.$$

Hence, we obtain the differential inequality

$$\frac{\mathrm{d}}{\mathrm{d}t}\left(\delta\mathbf{a}_t^\top \delta\mathbf{a}_t\right) \leq 2\lambda_{\max}(\mathbf{a}_t, t)\left(\delta\mathbf{a}_t^\top \delta\mathbf{a}_t\right),$$

and finally by applying Grönwall's inequality, this integrates to the contraction bound

$$\|\delta\mathbf{a}_t\| \leq \|\delta\mathbf{a}_0\|\exp\left(\int_0^t \lambda_{\max}(\mathbf{a}_\tau, \tau)\,\mathrm{d}\tau\right). \tag{17}$$

If $\lambda_{\max} < 0$, then any infinitesimal displacement $\delta\mathbf{a}_t$ decays exponentially to zero. Consequently, by integrating along any finite path, the total length contracts exponentially over time. □

**Terminal-to-initial sensitivity for reverse ODE.** From the proof of contraction above, and in particular the variational bound in Equation 17, we can view the *infinitesimal* statement as a *seed-sensitivity* bound between two reverse ODE solutions starting from different noise sampling. In other words, Equation 17 shows that contraction limits the action variance through:

Lower $\lambda_{\max} \to$ Lower $\|\delta\mathbf{a}_t\| \to$ Less sensitive to $\mathbf{a}_0$ sampling $\to$ Lower action variance.

Late in denoising, the signal to noise ratio gets higher, and the score field strongly points towards high-density regions which can be viewed as basins around dominant modes of conditional action distribution. Seeds started far apart often fall into the same basin for a fixed state given a fully trained score function. Contraction makes this basin narrower, rendering action generation more consistent, yet still capable of capturing multi-modal and complex behaviors.

### D.4 POWER ITERATIONS FULL STATEMENT

Let $A \in \mathbb{R}^{d \times d}$ be symmetric with eigen-decomposition $A = U\,\mathrm{diag}(\lambda_1, \ldots, \lambda_d)U^\top$ and orthonormal eigenvectors $U = [u_1, \ldots, u_d]$. Assume a *simple dominant eigenvalue in magnitude*:

$$|\lambda_1| > |\lambda_2| \geq \cdots \geq |\lambda_d|.$$

Choose any nonzero $v_0 \in \mathbb{R}^d$ with $\alpha_1 := u_1^\top v_0 \neq 0$. For example, draw $g \sim \mathcal{N}(0, I_d)$ and set $v_0 = g/\|g\|_2$; then $\alpha_1 \neq 0$ with probability 1. Define the power iteration and Rayleigh quotient by

$$v_{k+1} = \frac{Av_k}{\|Av_k\|_2}, \qquad \hat{\lambda}_k := v_k^\top Av_k, \quad k = 0, 1, 2, \ldots$$

---

[7] $q^\top = \left(\delta a^\top J^{\mathrm{asym}}\delta a\right)^\top = \delta a^\top J^{\mathrm{asym}\top}\delta a = \delta a^\top(-J^{\mathrm{asym}})\delta a = -\delta a^\top J^{\mathrm{asym}}\delta a = -q \xrightarrow{q \in \mathbb{R}} q = 0.$

Let $\theta_k := \angle(v_k, u_1)$. Then for all $k \geq 0$,

$$\tan \theta_k \leq \left( \frac{|\lambda_2|}{|\lambda_1|} \right)^k \frac{\sqrt{1 - \alpha_1^2}}{|\alpha_1|},$$

and the Rayleigh-quotient error satisfies

$$\left| \hat{\lambda}_k - \lambda_1 \right| \leq (|\lambda_1| + |\lambda_2|) \left( \frac{|\lambda_2|}{|\lambda_1|} \right)^{2k} \frac{1 - \alpha_1^2}{\alpha_1^2}.$$

Consequently, since $|\lambda_2|/|\lambda_1| < 1$, we have $\hat{\lambda}_k \to \lambda_1$ at a geometric (linear) rate, and $v_k$ converges to $\pm u_1$ at the same rate in angle (Austin et al., 2024).

### D.5 FROBENIUS NORM FOR PENALIZING THE LARGEST EIGENVALUE

**Frobenius surrogate with curvature shift.** Penalizing the Frobenius norm of a *shifted* symmetric Jacobian provides a simple, efficient surrogate for directly constraining its largest eigenvalue. For any matrix $J$,

$$\|J\|_F = \sqrt{\mathrm{tr}(J^\top J)} = \sqrt{\sum_i \mu_i^2}, \quad \|J\|_2 = \max_i \mu_i, \quad \|J\|_2 \leq \|J\|_F,$$

where $\{\mu_i\}$ are the singular values. When $J$ is symmetric, $\mu_i = |\lambda_i|$, so $\lambda_{\max}(J) \leq \|J\|_2 \leq \|J\|_F$. Rather than shrinking $J$ toward 0, we shrink it toward a *target curvature* $\beta \in \mathbb{R}^+$ by minimizing

$$\left\| J + \beta I \right\|_F^2 = \sum_i \left( \lambda_i(J) + \beta \right)^2,$$

which softly drives *all* eigenvalues toward $-\beta$.

**Application to diffusion ODE sampling.** Let $J_{\epsilon_\theta}^{\mathrm{sym}} = \frac{1}{2}(J_{\epsilon_\theta} + J_{\epsilon_\theta}^\top)$ denote the symmetric Jacobian of the (scaled) score field. Our shifted Frobenius penalty

$$\mathcal{L}_c^F(\theta) := \left\| J_{\epsilon_\theta}^{\mathrm{sym}} + \beta I \right\|_F$$

centers the regularization at $-\beta$. Indeed, for symmetric matrices,

$$\left| \lambda_{\max}(J_{\epsilon_\theta}^{\mathrm{sym}}) + \beta \right| \leq \left\| J_{\epsilon_\theta}^{\mathrm{sym}} + \beta I \right\|_F,$$

so shrinking the shifted Frobenius norm reduces an *upper bound* on the margin violation $\lambda_{\max}(J_{\epsilon_\theta}^{\mathrm{sym}}) + \beta$. Computationally, $\|\cdot\|_F$ is inexpensive to evaluate with automatic differentiation (sum of squares of entries), avoiding iterative eigensolvers.

*Remark.* The unshifted penalty $\|J_{\epsilon_\theta}^{\mathrm{sym}}\|_F$ only drives eigenvalues toward 0; a positive $\lambda_{\max}$ can persist while others move closer to 0, yielding a small Frobenius norm without satisfying contraction. In contrast, the shifted penalty $\|J_{\epsilon_\theta}^{\mathrm{sym}} - \beta I\|_F$ places the basin of attraction at the desired curvature $\beta$ (e.g., a negative margin), exerting greater pressure on eigenvalues that violate the target.

# E  COMPATIBILITY WITH DIFFUSION SOLVERS

In the main text, our focus has been on the ODE formulation of the sampling process. In this section, we illustrate with two prominent diffusion formulations, namely Denoising Diffusion Probabilistic Models (DDPM) (Ho et al., 2020) and Denoising Diffusion Implicit Models (DDIM) (Song et al., 2021a) that the Jacobian of the reverse process can be derived based on the score Jacobian and parameters of the forward process.

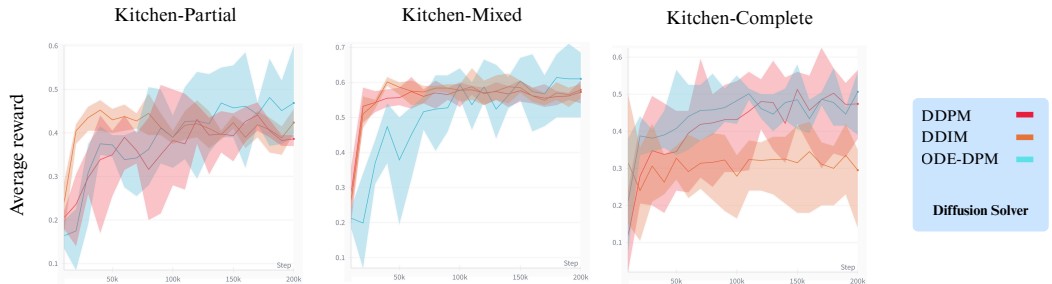

Figure 12: **Comparing Diffusion Solvers.** We change the diffusion solver of CDP to DDPM, DDIM, and the default ODE-based solver we use throughout this work. The average and error bands are plotted here.

## E.1  JACOBIAN IN THE DDPM FORMULATION

**DDPM overview.** DDPMs are essentially predecessors to diffusion SDE and ODE formulations. Similar to diffusion SDEs, they define a forward diffusion process where clean data $\mathbf{a}_0$ is corrupted into increasingly noisy versions $\mathbf{a}_1, \ldots, \mathbf{a}_T$ by adding Gaussian noise at each step, with carefully chosen variances so that $\mathbf{a}_T$ approaches pure noise. A neural network, usually parameterized to predict either the noise $\epsilon_\theta$ or the clean sample $\mathbf{a}_0$, is trained to approximate the reverse transitions: given a noisy input $\mathbf{a}_t$, predict how to denoise it toward $\mathbf{a}_{t-1}$.

**Jacobian derivation.** In the discrete-time and state-conditioned DDPM, the reverse step is

$$\mathbf{a}_{t-1} = \frac{1}{\sqrt{\alpha_t}}\left(\mathbf{a}_t - \frac{(1-\alpha_t)}{\sqrt{1-\bar{\alpha}_t}}\,\epsilon_\theta(\mathbf{a}_t,\,\mathbf{s},\,t)\right) + \sigma_t\,\xi, \qquad \xi \sim \mathcal{N}(0, I),$$

where $\bar{\alpha}_t = \prod_{s=1}^t \alpha_s$, and $\sigma_t^2 = \frac{1-\bar{\alpha}_{t-1}}{1-\bar{\alpha}_t}(1-\alpha_t)$. Since $\xi$ is independent of $\mathbf{a}_t$ and $\mathbf{s}$ is fixed, they do not contribute to the Jacobian. The per-step Jacobian of the reverse mapping can therefore be derived from the score Jacobian as

$$J_{\theta,t} \triangleq \frac{\partial \mathbf{a}_{t-1}}{\partial \mathbf{a}_t} = \frac{1}{\sqrt{\alpha_t}}\left(I - \frac{(1-\alpha_t)}{\sqrt{1-\bar{\alpha}_t}}\frac{\partial \epsilon_\theta(\mathbf{a}_t,\,\mathbf{s},\,t)}{\partial \mathbf{a}_t}\right).$$

Thus, by repeated application of the chain rule across the sequence of reverse steps, the Jacobian of the entire sampling process (conditioned on $\mathbf{s}$) is expressed as the product of the per-step Jacobians:

$$J_\theta = \prod_{t=T}^1 J_{\theta,t}.$$

Note that even though DDPM is written as a discrete reverse map, it can be viewed as a time-discretization of a continuous reverse flow ODE (Ho et al., 2020; Song et al., 2021b).

## E.2  JACOBIAN IN THE DDIM FORMULATION

**DDIM overview.** DDIMs provide a deterministic alternative to DDPM sampling by reinterpreting the reverse process as a non-Markovian chain. Instead of treating the reverse diffusion as a stochastic

step with added Gaussian noise, DDIMs define a deterministic mapping from $\mathbf{a}_t$ to $\mathbf{a}_{t-1}$ through a reparameterization that directly leverages the network prediction of $\epsilon$ or $\hat{\mathbf{a}}_0$. This eliminates the sampling noise while preserving the same marginal distributions as DDPMs, thereby allowing faster generation with fewer steps and enabling more controllable trajectories.

**Jacobian derivation.** Let $\bar{\alpha}_t \in (0, 1]$ with $\sigma_t \coloneqq \sqrt{1 - \bar{\alpha}_t}$. The deterministic DDIM variance-preserving update is

$$\mathbf{a}_{t-1} = \sqrt{\bar{\alpha}_{t-1}} \, \hat{\mathbf{a}}_0(\mathbf{a}_t, \mathbf{s}, t) + \sqrt{1 - \bar{\alpha}_{t-1}} \, \epsilon_\theta(\mathbf{a}_t, \mathbf{s}, t), \tag{18}$$

where the predicted clean sample $\hat{\mathbf{a}}_0$ can be also written based on the score function as

$$\hat{\mathbf{a}}_0(\mathbf{a}_t, \mathbf{s}, t) = \frac{1}{\sqrt{\bar{\alpha}_t}} \Big( \mathbf{a}_t - \sqrt{1 - \bar{\alpha}_t} \, \hat{\varepsilon}_\theta(\mathbf{a}_t, \mathbf{s}, t) \Big). \tag{19}$$

After replacing Equation 19 into Equation 18, it becomes clear that we can essentially repeat the same process as DDPM and find the Jacobian of the entire process.

## F ADDITIONAL EXPERIMENTS

### F.1 OFFLINE RL AND IL EXPERIMENTS

In Figure 14, we display the evaluation graphs for CDP with two different contraction rates, compared against EDP (Kang et al., 2023), which employs diffusion sampling without any notion of contraction. Despite a simplistic vector search for contraction hyperparameters (Section 4), CDP consistently improves performance across many environments, while achieving comparable or only slightly lower rewards in others. The only notable *exceptions* are HalfCheetah-M and Walker2D-ME, where CDP underperforms. We attribute this primarily to the limited scope of our hyperparameter search. Yet, other factors such as the choice of diffusion solver or the number of sampling steps may contribute and warrant further investigation in future work.

Figure 16 and Figure 17 present the average evaluation reward for the Robomimic-Lowdim and Robomimic-Image environments, respectively. For Robomimic-Lowdim tasks, we report results under three settings: limited data, noisy actions, and the standard training process. Across all experimental settings, CDP demonstrates a consistent improvement in average reward over the baseline, with particularly strong gains in the more challenging limited-data and noisy-action regimes. These findings mirror our observations on D4RL: the benefits of contractive regularization become more pronounced in scenarios where the diffusion process is highly susceptible to score-matching inaccuracies and solver discretization errors, such as when data is scarce, actions are noisy, or learning relies on high-dimensional image inputs.

### F.2 EFFECT OF DIFFERENT CONTRACTION LOSS WEIGHTS

We examine how different contraction loss weights influence evaluation performance across both RL and IL environments, with results summarized in Figure 15. Overall, we observe a general trend in which overly strong enforcement of contraction leads to a drop in performance. To provide additional intuition, Figure 18 presents a toy diffusion modeling example using four log-spaced contraction weights. The resulting distributions of sampled actions across training epochs clearly illustrate how contraction shapes the diffusion sampling process. Lastly, we portray how various contraction weights change the largest eigenvalue during training to better showcase the way the contraction loss affects the model training, especially the decreasing effect on the larges eigen value.

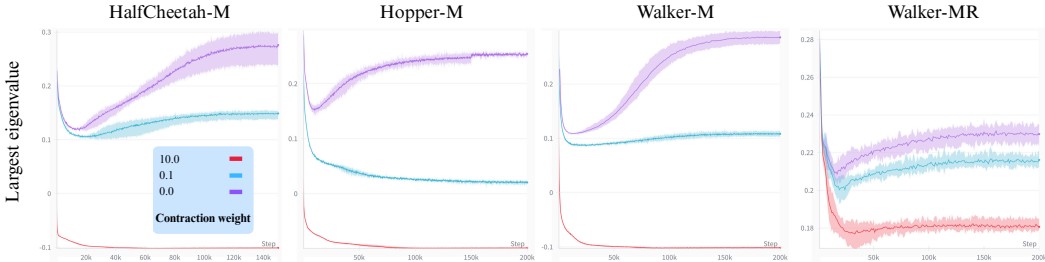

Figure 13: **Largest eigenvalue with contraction.** Higher rates of contraction naturally push the largest eigenvalues (along with the rest) to be more negative.

### F.3 CDP WITH FEWER SOLVER STEPS

One of the central claims in the main text is that contraction dampens score-matching and solver-related errors, including discretization errors and inaccuracies in score estimation arising in limited-data scenarios. Figure 19 illustrates this effect by comparing samples from contractive and standard diffusion processes at various training epochs under a reduced number of solver steps. Reducing the number of solver steps forces each integration step to cover a larger interval, which amplifies discretization error and makes the reverse process less faithful to the underlying diffusion dynamics. As a result, the quality of the generated samples typically degrades when fewer steps are used. Nevertheless, the plots show that CDP remains more robust in this setting, producing samples that are consistently closer to the ground-truth actions, thereby demonstrating its ability to mitigate solver-induced inaccuracies.

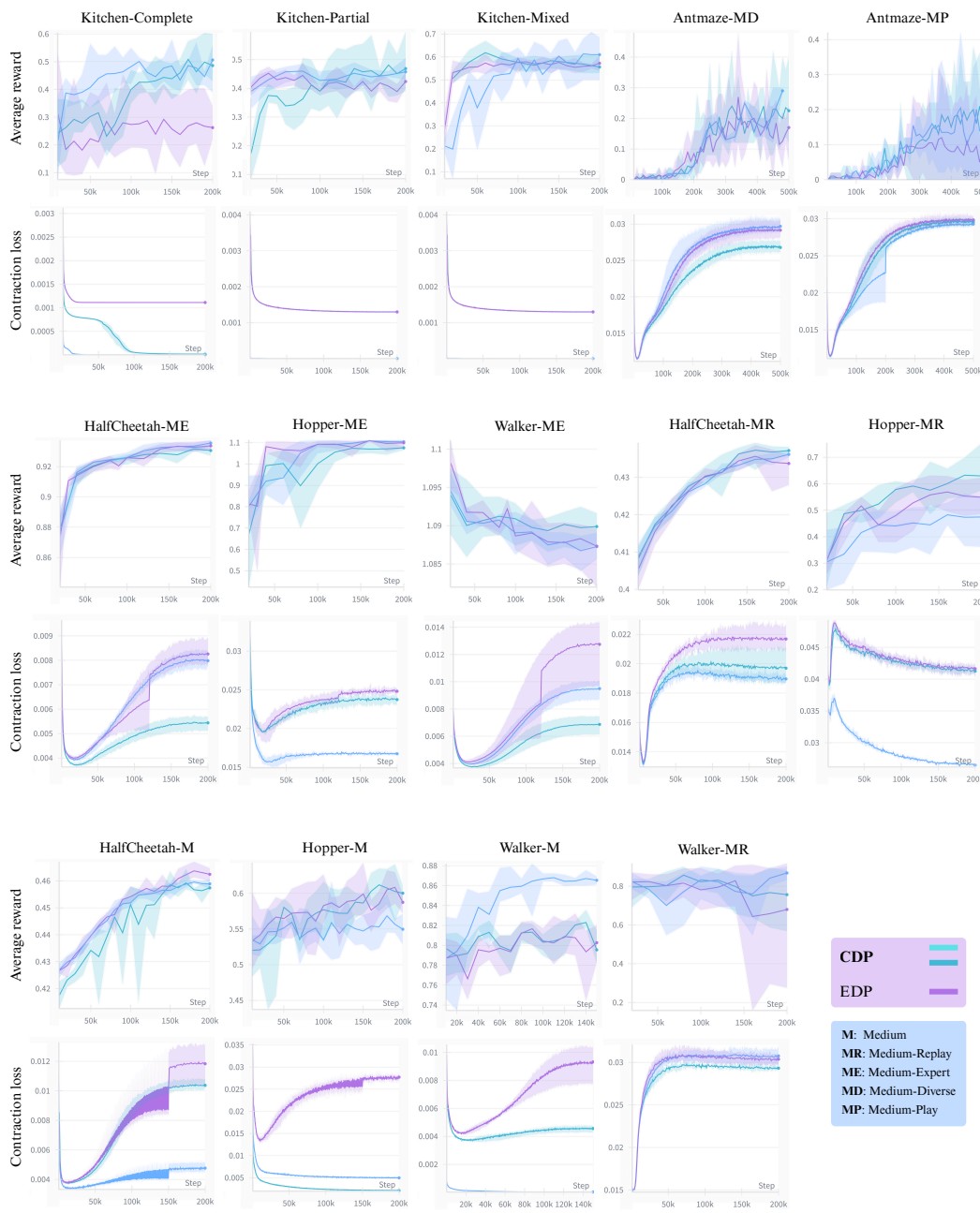

Figure 14: **D4RL evaluation reward and contraction loss graphs.** Average evaluation reward and contraction loss are reported across a range of D4RL environments and tasks. We compare CDP under different contraction loss coefficients—applied using the *Jacobian loss* formulation—against the baseline EDP method. To highlight the effect of enforcing contractivity, we also compute the contraction loss for both EDP and CDP rollouts, which clearly illustrates the gap between methods trained with and without contraction. Note that the contraction loss is measured for EDP but *not penalized*. Finally, we observe that contraction loss typically drops sharply at the beginning of training, but then rebounds and stabilizes as minimizing the diffusion loss becomes impossible with extreme levels of contraction. We plot the error bands in addition to average.

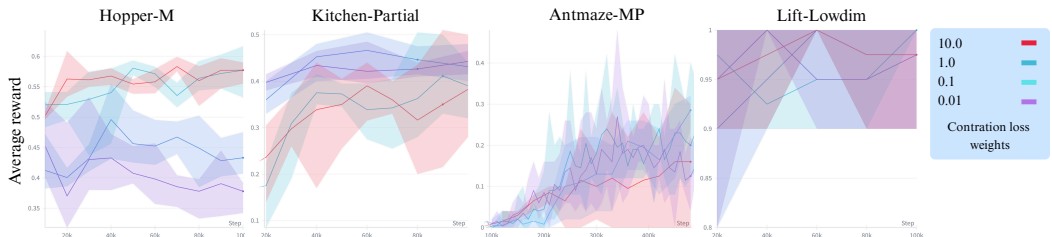

Figure 15: **Performance with changing contraction rates.** We report evaluation curves and error bands for four tasks and environments under varying contraction loss weights $\gamma$. All other hyperparameters are held fixed across runs, and for each setting we plot the mean and variance of evaluation performance over 10 random seeds.

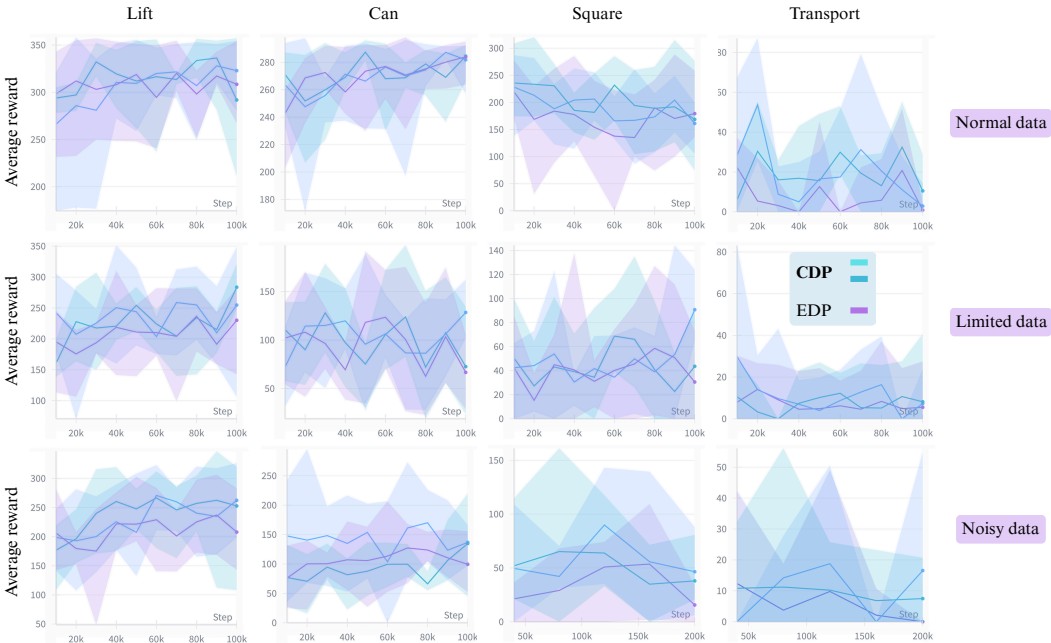

Figure 16: **Robomimic-Lowdim evaluation reward graphs.** We evaluate on the standard Robomimic datasets, as well as two perturbed settings: a limited-data scenario containing only 10% of the original samples, and a noisy dataset constructed by adding Gaussian noise with average magnitude 0.1 to the actions. While the performance gains in simpler environments with standard data are modest, more challenging tasks and the limited/noisy settings highlight clearer advantages. We plot the error bands in addition to average.

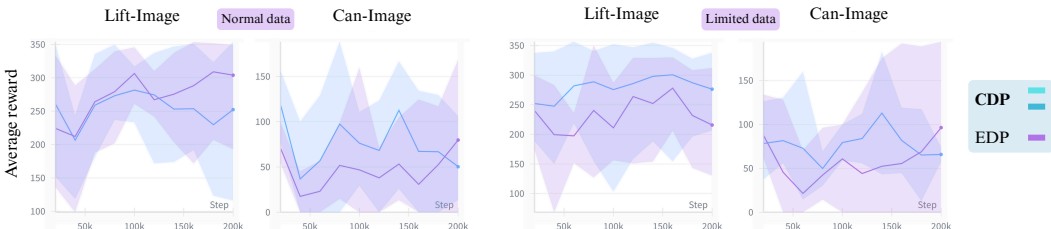

Figure 17: **Robomimic-Image evaluation reward graphs.** In these environments, the observation space consists of two image modalities from the agentview and wrist cameras, augmented with low-dimensional proprioceptive data. Due to computational constraints, we restrict experiments to the Lift and Can tasks. The results suggest that CDP is particularly effective when learning from visual inputs; however, we defer a more thorough investigation of this setting to future work.

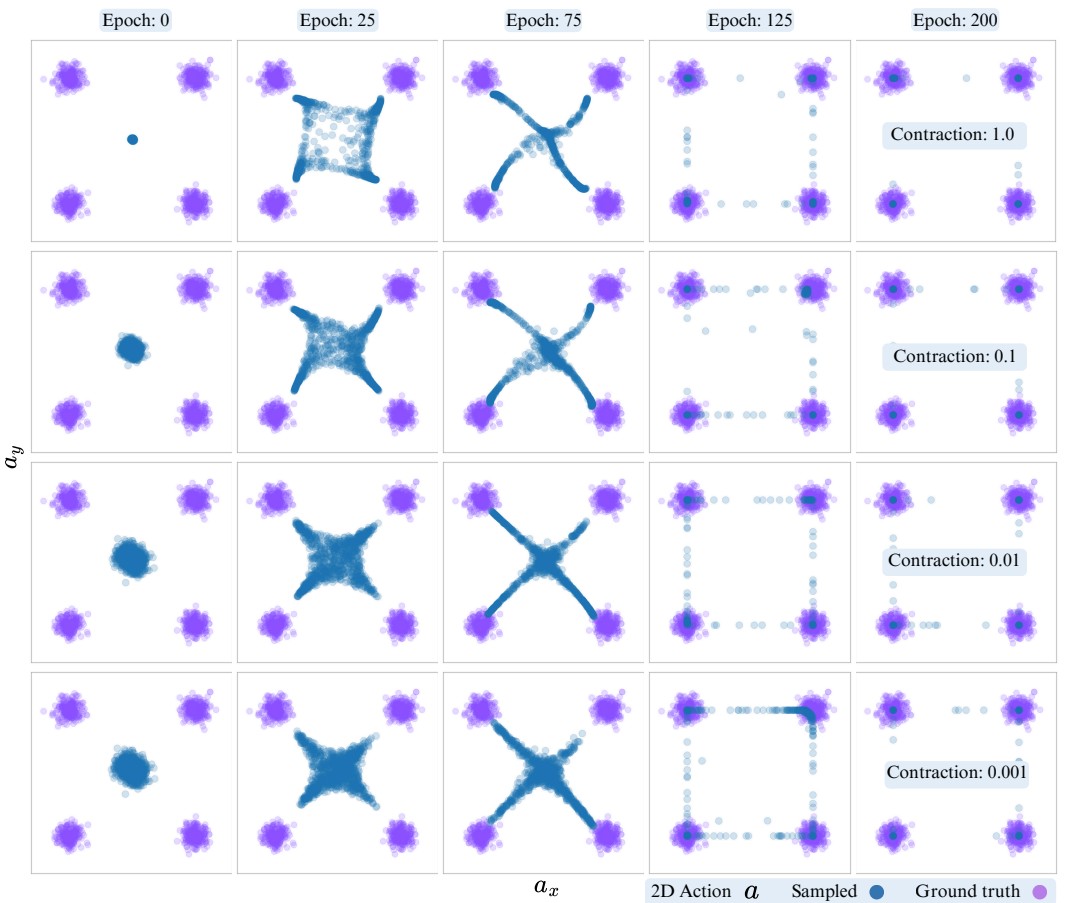

Figure 18: **CDP sampling for various contraction loss weights.** The comparison illustrates the way contraction influences both the diffusion dynamics during training and the final policy performance. In this example, however, the difference between setting $\gamma = 0.1$ and increasing it by an order of magnitude is negligible in practice. Across similar experiments, no universal rule of thumb is identified, but logarithmic hyperparameter search provides a practical and effective strategy.

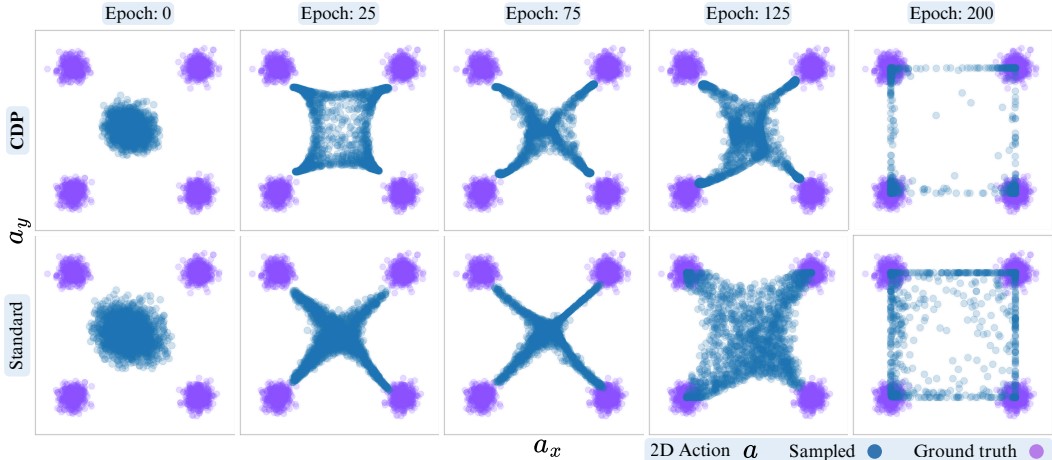

Figure 19: **Diffusion sampling with fewer steps.** Reducing the number of sampling steps forces the solver to take larger updates, which decreases accuracy and amplifies discretization error. In later training epochs, we observe that contraction helps dampen these solver inaccuracies and stabilizes the sampling process; however, it cannot fully eliminate the errors introduced by coarse discretization.

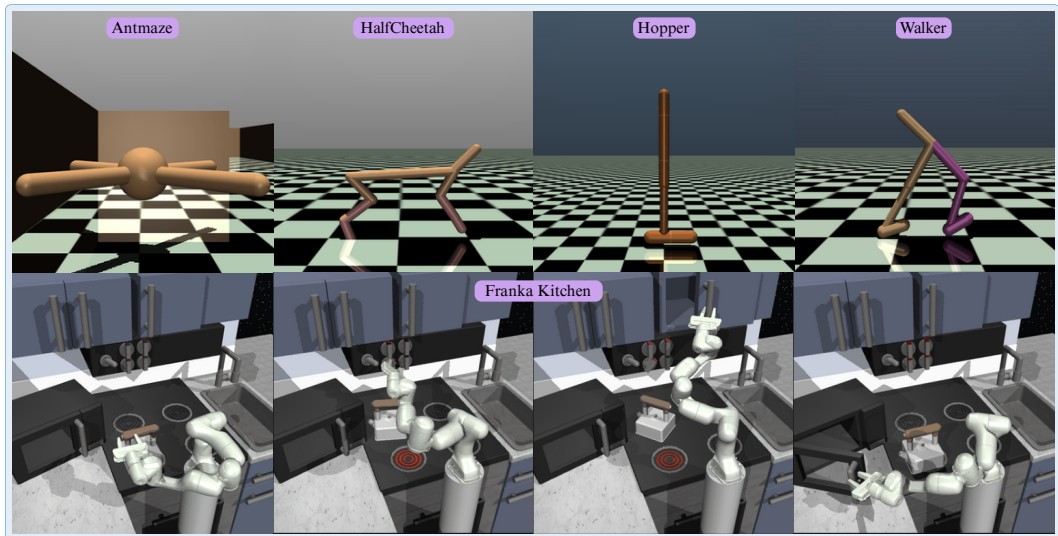

Figure 20: **D4RL environments.** These environments are used for offline RL benchmarking.

## G DATASETS AND BENCHMARKS

This section provides a technical overview of the D4RL and Robomimic datasets used in our experiments. We detail the tasks, the nature of the offline data, and the evaluation protocols for policies trained on these datasets.

### G.1 D4RL: DATASETS FOR DEEP OFFLINE REINFORCEMENT LEARNING

The D4RL benchmark (Fu et al., 2020) is a collection of standardized tasks and static datasets designed to facilitate reproducible research in offline RL and IL. The core challenge is to learn effective policies from a fixed batch of data without further interaction with the environment during training. Once trained, policies are evaluated in their corresponding simulated environments. An overview of the environments used in our study is shown in Figure 20.

**Franka Kitchen**. This task involves controlling a 9-DoF Franka robot arm to perform a sequence of manipulation tasks in a kitchen setting. It is designed to test an agent's ability to learn long-horizon behaviors from complex, unstructured data. We use the following datasets:

- `complete`: Trajectories demonstrate the successful completion of all required subtasks.
- `partial`: Trajectories include the target subtasks, but also contain unrelated actions. An agent must identify and imitate the relevant segments.
- `mixed`: No single trajectory completes the full task. The agent must learn to "stitch" together sub-trajectories from different demonstrations to succeed.

**Gym-MuJoCo**. These are classic locomotion tasks where the goal is to control agents to move forward as quickly as possible. We focus on the Half-Cheetah, Hopper, and Walker-2D environments. The datasets are generated from policies at varying levels of proficiency:

- `medium`: Data collected from a policy trained to a moderate level of performance.
- `medium-replay`: The entire replay buffer content from the training process up to the `medium` policy level. This dataset is larger and more diverse.
- `medium-expert`: A dataset composed of 50% `medium` data and 50% data from a near-optimal `expert` policy.

**AntMaze**. In this challenging navigation task, an 8-DoF "Ant" agent must navigate a maze to a target location. The reward is sparse (a positive reward is given only upon reaching the goal),

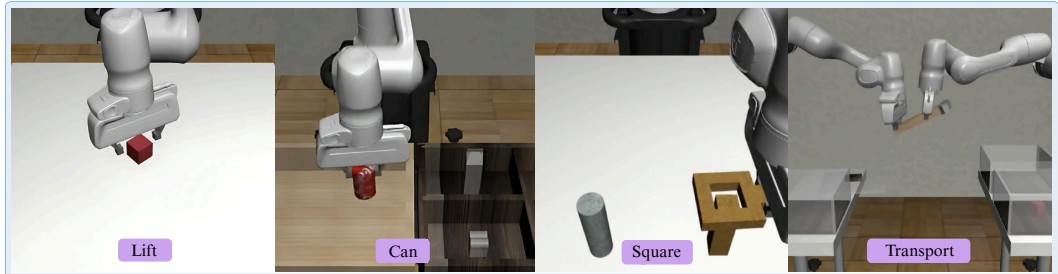

Figure 21: **Robomimic tasks.** These tasks are used in our study for IL benchmarking in robosuite.

requiring the agent to learn from long, often unsuccessful, trajectories. The datasets we use are `medium-diverse` and `medium-play`, which feature trajectories from a moderately skilled policy attempting to reach various goals within the maze.

**Evaluation Protocol**. For all D4RL tasks, the trained policy is evaluated by deploying it in the corresponding MuJoCo-based simulator. Performance is measured by the average cumulative reward over several evaluation episodes. For the AntMaze tasks, this corresponds to the success rate of reaching the goal.

### G.2 ROBOMIMIC: LEARNING FROM HUMAN DEMONSTRATIONS

Robomimic (Mandlekar et al., 2021) is a large-scale benchmark for IL, featuring datasets collected from human teleoperation. The tasks are simulated in the `robosuite` framework (Zhu et al., 2020). Learning from this data is particularly challenging because human demonstrations can be non-Markovian, inconsistent, and of varying quality. The tasks used in our experiments are visualized in Figure 21.

**Manipulation Tasks**. We evaluate our methods on four tabletop manipulation tasks:

- **Lift**: Pick up a cube from a table.
- **Can**: Pick up a can and place it into a bin.
- **Square**: Pick up a cube and place it inside a designated square region.
- **Transport**: Pick up an object from one bin and move it to another.

**Evaluation Protocol**. Policies are trained on the provided human demonstration data and then evaluated in the `robosuite` simulator. The primary metric for all Robomimic tasks is the **success rate**, defined as the percentage of episodes where the agent successfully completes the task objective. The final performance is reported as the mean success rate over 100 evaluation rollouts.

### G.3 REAL WORLD DATA WITH FRANKA ROBOT

Information about tasks and how the demonstrations are collected is provided in Appendix B. Here, we provide a general profile of the dataset sizes in Table 5.

Table 5: **Characteristics of the collected real-world datasets.** Observations consist of agent-view RGB images and proprioceptive states (end-effector position and orientation in `xyz-rpy`), while actions correspond to delta end-effector motions in the six-dimensional `xyz-rpy`.

| Task | # Demonstrations | Trajectory Length (min–max) | Max Length | Size (MB) |
|---|---|---|---|---|
| Slide (cylinder) | 51 | 148–209 | 209 | 7989.45 |
| Stack (cubes) | 41 | 182–286 | 286 | 7908.29 |
| Lift (cube) | 31 | 107–224 | 224 | 3782.33 |
| Peg (insert in hole) | 61 | 97–296 | 296 | 8976.17 |

