# OpenReview forum: "Contractive Diffusion Policies"
_ICLR.cc/2026/Conference — ICLR 2026 Poster_

### Official Review · Reviewer_T4PE · 2025-11-01

**Soundness:** 3
**Presentation:** 3
**Contribution:** 3
**Rating:** 6
**Confidence:** 4

**Summary:**

This paper identifies a weakness in modern diffusion policies. The authors argue that the iterative nature of the diffusion sampling process introduces compounding errors from two sources 1) inaccuracy of the learned score function, especially in low-data cases, and 2) discretization errors from the ODE solver. In control and robotics, these small and accumulating errors can lead to failed actions.

To solve this, the authors introduce Contrastive Diffusion Policy (CDPs). The core idea is to leverage contraction theory to make the reverse diffusion ODE more stable. The contraction theory establishes the relation between the ODE's Jacobian and the ODE. If this condition is satisfied, it can guarantee that the ODE trajectories converge to a more concentrated target distribution. This property makes the learned ODE more concentrated and the sampling process inherently robust to small perturbations like solver or model errors.

This paper leverages the theoretical analysis of contraction theory for ODE and modifies the original diffusion or flow policy by forcing the ODE's Jacobian to satisfy the contrastive drifting condition. This is learned along with the original flow objective. For the training details, they use the power iteration method to approximate the largest eigenvalue of the score Jacobian. Then a contraction loss is applied to penalize the model to be more concentrated. The final training loss is a simple weighted sum of the original diffusion loss and the new contraction loss.

Experiments on D4RL, Robomimic, and real-world Franka robot tasks demonstrate that CDPs outperform the standard diffusion policy baseline, with particularly strong performance in low-data scenarios.

**Strengths:**

- The paper leverages a solid foundation in contraction theory and gives a formal analysis of the ODE's stability, which is insightful and provides a clear, principled target for regularization
- The paper has a good motivation. Compounding error is a valid issue in continuous control
- The paper includes physical robot experiments in the real world, which demonstrates its robustness in practice

**Weaknesses:**

- The paper describes the added cost as "minimal" and "negligible." However, the proposed method requires computing a Jacobian-vector product (K=3 or 4 times for the power iteration) for every item in the batch at every single training step. This is definitely more computationally expensive than a standard diffusion loss. A more transparent analysis (e.g., a wall-clock time comparison) would be helpful to quantify this overhead.
- The advantage of diffusion models is to capture complex, multi-modal distributions (like the different ways to perform a task). Forcing the system to be "contractive" (pulling trajectories together) seems intuitively to contradict preserving this diversity. The paper mentions that "excessive penalization could fuel a mode collapse," but does not investigate this trade-off in-depth. A quantitative study on how $\gamma$ affects the variance or modality of the final action distribution would make the paper more complete, especially on a task showing multi-modal behavior.
- The authors acknowledge this limitation of hyperparameter sensitivity. The new contraction loss weight, $\gamma$, is a critical hyperparameter. As shown in Table 4, the optimal value for $\gamma$ varies dramatically across different environments (from 0.001 to 100.0). This high sensitivity could make the method difficult to apply to new tasks without a costly hyperparameter sweep

**Questions:**

In addition to the questions in the weakness above, I have the following additional questions

- Regarding Stochastic (SDE) Solvers: The paper focuses on stabilizing the deterministic ODE sampling process (and the experiments appear to use the deterministic ODE sampler). However, stochastic samplers (like DDPM) are well-known to have error-correcting properties due to their Langevin noise term, which also tends to concentrate the final distribution.
Could the authors comment on why an ODE-based solver was chosen over an SDE-based one?
Did the authors experiment with stochastic samplers? If so, how do they perform? Does the inherent noise of an SDE sampler already provide a similar "contractive" effect, making the explicit contraction loss less necessary?

- Regarding Simpler Alternatives (e.g., Data Re-weighting): As an alternative to regularizing the model, could a similar "concentrating" effect be achieved through the training data or loss function? For example, one could identify actions that are "central" or "high-quality" (e.g., near the mean of a mode in the data) and apply a higher loss weight to these samples during training. This would intuitively "tilt" the learned score function to flow towards these more robust actions, without the computational cost of Jacobian-vector products. Could the authors comment on the pros and cons of their proposed method versus such a data re-weighting or distribution tilting approach?

- In Table 2 (Robomimic), the proposed CDP (0.78 avg) underperforms the DP-Unet baseline (0.88 avg) significantly. Could the author provide additional results with CDP + Unet to show the improvements over different architectures?

- I'm not quite sure why the method performs better for low-data regimes. if the score field ε_θ is inaccurate due to sparse data, why does forcing its Jacobian to be contractive (based on this inaccurate field) lead to a better outcome? Is there a theoretical explanation for this?

*Typo: line 246 J_{\epsilon_\theta}^{sym} seems undefined?

---

> ### Author Response · Authors · 2025-11-19
> **Response to Reviewer T4PE**
>
> We are grateful for the reviewer’s constructive and detailed feedback. We have carefully addressed all the comments in the order they were raised and made further refinements to the paper. An updated version of the manuscript is released with changes in red.
> ```
> 1. The paper describes the added cost as "minimal" and "negligible." However, the proposed method requires computing a Jacobian-vector product (K=3 or 4 times for the power iteration) for every item in the batch at every single training step. This is definitely more computationally expensive than a standard diffusion loss. A more transparent analysis (e.g., a wall-clock time comparison) would be helpful to quantify this overhead.
> ```
>
> Our method is indeed more computationally intensive than standard diffusion. Yet, we observed that unlike regular Jacobian computation, our contraction loss only requires the largest eigenvalue, which we effectively compute with power iterations in Sec. 3.2, and compare its efficiency in Fig. 11, which shows a remarkable boost. Yet, as you pointed out, it does not compare against standard diffusion.
>
> To address this concern, **we updated Tables 1 and 2 in our paper to include a proper time comparison with standard diffusion policies, since we always log the execution times.** The results show a reasonable overhead of roughly 9% for imitation learning tasks (compared to DBC-DiT) and 14% for reinforcement learning (compared to EDP), noting that CDP is presently built on top of EDP and DBC. In addition, **we have updated the manuscript** to more precisely characterize the computational overhead, avoiding the term negligible since this could be subjective based on available compute.
>
> ```
> 2. The advantage of diffusion models is to capture complex, multi-modal distributions (like the different ways to perform a task). Forcing the system to be “contractive” (pulling trajectories together) seems intuitively to contradict preserving this diversity. The paper mentions that “excessive penalization could fuel a mode collapse” but does not investigate this trade-off in-depth. A quantitative study on how $\gamma$ affects the variance or modality of the final action distribution would make the paper more complete, especially on a task showing multi-modal behavior.
> ```
>
> We understand that the “phrase pulling trajectories” together seems contradictory with multimodal learning. That is why we use the phrase “nearby flows or trajectories” in most places to highlight that contraction can be used to locally pull nearby flows while keeping the main modes of data untouched. **We updated the paper, especially Section 3.2 to better reflect this.**
>
> To elaborate, extreme contraction levels applied globally can indeed be harmful to the multimodal nature of diffusion models, and by formulating a conservative contraction loss, we make sure our training objective avoids such scenarios. The contraction loss imposed in our method is applied only at the Jacobian of the score function, which is regularized to enforce local stability. This pulls nearby trajectories together to reduce variance and improve robustness, but it does not collapse distinct, well-separated modes. Therefore, preserving multimodality by learning an accurate and diverse score remains a critical objective as long as over-regularization is avoided and the diffusion loss remains critical.
>
> **We added Appendix C.6 to more clearly highlight the measures we provide to make over-regularization unlikely.** In particular, we explain that we mitigate such cases by using a Frobenius-norm surrogate with a conservative threshold ($\beta$ in the paper) and by enforcing contraction primarily at the latest stages of the denoising process (contr_steps in Appendix C.4). However, since we set these parameters conservatively, we only tune the performance using the contraction regularization parameter $\gamma$ and treat it as the only critical hyperparameter of our approach.
>
> These measures pay off in practice, where we do not observe a measurable collapse in our experiment over a range of regularization coefficients: CDP improves or matches baselines on most D4RL and Robomimic (compared to DBC) tasks (Tables 1–2), with especially clear gains in low-data regimes (Fig. 5) and on real hardware (Fig. 6). Importantly, we keep the contraction threshold fixed across all experiments (i.e., we do not tune it per task). Finally, despite theoretical differences, prior work [6] (in particular Section 4, Theorem 7) provides additional evidence that contraction can improve the process without damaging the flexibility of the diffusion process.

---

> ### Author Response · Authors · 2025-11-19
> **Response to Reviewer T4PE -- Continued**
>
> ```
> 3. The authors acknowledge this limitation of hyperparameter sensitivity. The new contraction loss weight, $\gamma$, is a critical hyperparameter. As shown in Table 4, the optimal value for $\gamma$ varies dramatically across different environments (from 0.001 to 100.0). This high sensitivity could make the method difficult to apply to new tasks without a costly hyperparameter sweep.
> ```
>
>  While $\gamma$ requires tuning, we find a coarse search is sufficient. However, we showcase that our method benefits from contraction with naive and straightforward hyperparameter tuning. Although the effective scale of the contraction coefficient depends on factors like the action distribution, demo quality, and sample size. In practice, we deliberately use a minimal, six-value grid of [0.001, 0.01, 0.1 ... 100] (Sec. 4.1) for all tasks and benchmarks, while many methods in the field have to change the hyperparameter search range per task [1, 2]. Despite the variation in Table 4, this coarse, low-cost sweep consistently exposes useful (\gamma) settings and yields clear gains, which suggests the method is deployable without expensive tuning. Such evidence establishes the contraction coefficient as a low-cost and beneficial hyperparameter.
>
> ```
> Q1. Regarding Stochastic (SDE) Solvers: The paper focuses on stabilizing the deterministic ODE sampling process (and the experiments appear to use the deterministic ODE sampler). However, stochastic samplers (like DDPM) are well-known to have error-correcting properties due to their Langevin noise term, which also tends to concentrate the final distribution. Could the authors comment on why an ODE-based solver was chosen over an SDE-based one? Did the authors experiment with stochastic samplers? If so, how do they perform? Does the inherent noise of an SDE sampler already provide a similar "contractive" effect, making the explicit contraction loss less necessary?
> ```
>
> The choice of an ODE-based solver is motivated by two practical considerations. First, the original diffusion literature and multiple follow-up works show that ODE and SDE solvers are *equivalent* in the sense that they produce the same underlying distribution (see Theorem 2.1 in our paper and [4]). Second, the contraction-theoretic formulation is significantly more tractable in the ODE setting than in the SDE case [3].
>
> As part of the revised paper, **we have included additional results (Figure 12, Appendix E) comparing DDPM, DDIM, and ODE-based solvers on Franka Kitchen [2], and we observe comparable performance across all methods.** Yet, we have not yet conducted extensive large-scale experiments with SDE solvers, given that the equivalence to ODE solvers is well-established [4]. However, if the reviewer believes that explicit SDE results would strengthen the paper, we would be happy to include them in the appendix.
>
> It is worth noting that, theoretically, while Langevin noise in SDE/ancestral samplers might correct local errors and sometimes concentrate mass, it does not enforce a negative symmetric Jacobian or cancel systematic score/discretization bias [5].
>
> ```
> Q2. Regarding Simpler Alternatives (e.g., Data Re-weighting): As an alternative to regularizing the model, could a similar "concentrating" effect be achieved through the training data or loss function? For example, one could identify actions that are "central" or "high-quality" (e.g., near the mean of a mode in the data) and apply a higher loss weight to these samples during training. This would intuitively "tilt" the learned score function to flow towards these more robust actions, without the computational cost of Jacobian-vector products. Could the authors comment on the pros and cons of their proposed method versus such a data re-weighting or distribution tilting approach?
> ```
>
> The proposed approach is interesting. Tilting the data or loss toward central/high-quality actions can indeed increase robustness with potentially less computational complexity than what we propose, but it tackles a different failure mode than ours. Re-weighting reshapes the training distribution and thus the score estimate itself; it can bias the model toward means/modes that look central under the chosen heuristic, but risks suppressing legitimate tails and multi-modal structure, and most importantly, it hinges on reliably measuring per state (hard in robotics, often domain-specific, and could appear brittle under covariate shift).
>
> By contrast, our contraction loss does not change what the model should represent. It controls the local sampling sensitivity of the reverse dynamics at test time (especially in late denoising), curbing amplification of score/solver errors while leaving data-consistent multimodality intact. So in this way, we can improve the performance without any specific and reliable measure or metric, and without curating the dataset samples.

---

> ### Author Response · Authors · 2025-11-19
> **Response to Reviewer T4PE -- Continued**
>
> ```
> 3. In Table 2 (Robomimic), the proposed CDP (0.78 avg) underperforms the DP-Unet baseline (0.88 avg) significantly. Could the author provide additional results with CDP + Unet to show the improvements over different architectures?
> ```
>
> We acknowledge this issue. We tried to explain it in the paper, Sec. 4.2, Q1. To phrase it better, DP in general performs DBC approaches, and we build CDP on top of a DBC implementation. To address this concern, **we ran more experiments with CDP implemented directly on the DP architecture (through a simple change of diffusion model in our codebase), and added the results to Table 2.** These experiments show an overall advantage over the DBC-based implementations, particularly for more complex vision tasks. Note that we ran for 5 seeds presently to get the results faster, but will add 5 more seeds in the coming days.
>
> ```
> 4. I'm not quite sure why the method performs better for low-data regimes. If the score field ε_θ is inaccurate due to sparse data, why does forcing its Jacobian to be contractive (based on this inaccurate field) lead to a better outcome? Is there a theoretical explanation for this?
> ```
>
> We cannot claim that contraction fixes an inaccurate score, rather, it limits how errors propagate through the sampler. In essence, if the score function is inaccurate at every step, contraction cannot help. However, if the score is inaccurate for only some denoising steps, a situation that arises when we lower the number of data points, contraction could help control the resulting error propagation. Hence, the dataset is small enough to cause inaccuracy for some denoising steps but not major mode failures.
>
> For a deeper dive, let $F_\theta(t,x)=f(t)x+h(t)\epsilon_\theta(t,x)$ be the reverse field and $F^\star$ its ideal counterpart. The terminal action error $|x_T^\theta-x_T^\star|$ depends on both the score error $e=\epsilon_\theta-\epsilon^\star$ and the flow’s sensitivity, which is controlled by the largest eigenvalue of the symmetric Jacobian:
>
>  $$|x_T^\theta-x_T^\star| \lesssim \exp \Big(\int_0^T \lambda_{\max} \big(J^{\text{sym}}({F_\theta})\big)dt\Big)\cdot \int_0^T |h(t)e(t,x_t)|dt.$$
>
> In low-data regimes, $|e|$ is naturally larger. Without contraction, the exponential factor can amplify these errors (along with numerical integration errors) into large action deviations. Enforcing a negative margin on $\lambda_{\max}(J^{\text{sym}}{F_\theta})$ slows down the growth, so both seed noise and model/solver errors decay along the reverse trajectory. In short, contraction is a stability prior that trades a small bias for a large reduction in error amplification, which is precisely where low-data models suffer.
>
> ```
> 5. *Typo: line 246 J_{\epsilon_\theta}^{sym} seems undefined?
> ```
>
> Thank you for noticing this issue. The sym notation denotes the symmetric part of a matrix. **We have updated the paper accordingly.**
>
> Thank you again, and we hope to have sufficiently addressed your concerns. Nonetheless, we are here for additional discussion, and to provide more results to help strengthen the work.
>
> **References**
>
> [1] Park S, Li Q, Levine S. Flow q-learning. arXiv preprint arXiv:2502.02538. 2025 Feb 4.
>
> [2] Fujimoto S, Gu SS. A minimalist approach to offline reinforcement learning. Advances in neural information processing systems. 2021 Dec 6;34:20132-45.
>
> [3] Tsukamoto H, Chung SJ, Slotine JJ. Contraction theory for nonlinear stability analysis and learning-based control: A tutorial overview. Annual Reviews in Control. 2021 Jan 1;52:135-69.
>
> [4] Song Y, Sohl-Dickstein J, Kingma DP, Kumar A, Ermon S, Poole B. Score-Based Generative Modeling through Stochastic Differential Equations. In International Conference on Learning Representations.
>
> [5] Dockhorn T, Vahdat A, Kreis K. Score-based generative modeling with critically-damped langevin diffusion. arXiv preprint arXiv:2112.07068. 2021 Dec 14.

---

### Official Review · Reviewer_rXr6 · 2025-11-03

**Soundness:** 3
**Presentation:** 4
**Contribution:** 2
**Rating:** 2
**Confidence:** 3

**Summary:**

The paper is improving general diffusion model policies through an additional class of losses that promote contractions in the sampling processes of the diffusion model policy. These losses are denoted Contractive Diffusion Policies (CDPs), which add a Jacobian-based loss so the reverse diffusion dynamics are contractive. The goal of this is to damp solver and score noise and reduce action variance. Experiments are done on D4RL, Robomimic, and a small real-robot setup.

**Strengths:**

- Simple idea that integrates into standard diffusion-policy training without major architectural changes.
- Well-presented theory around contraction for ODE’s and diffusion (or more generally flow-based) generative models.
- Some positive results on D4RL benchmarks, especially where reducing variance helps.

**Weaknesses:**

- Contraction seems to fundamentally contradict a central reason practitioners use diffusion model polices over e.g. Gaussian policies, namely multimodality. By pulling trajectories together, CDP can collapse valid modes that diffusion policies are meant to capture. It then seems in this setting that Gaussian policies would need to be benchmarked against, or at the very least explanation provided as to why either (a) multimodality is actually not that important for RL policies, or (b) why the contraction loss does not actually eliminate multimodality when it’s efficient for optimal policies to inhibit this.


- There is no direct evidence that the spectral condition for contraction actually holds during training and sampling. Plots of eigenvalues through various time conditioning would give future readers a better idea of the effect of the contraction loss, and what parts of the diffusion model become more strongly contractive.

- As Jacobian computations are typically expensive, a computation time analysis here seems relevant.


- While some environments show CDP giving a non-negligible boost, none of the robomimic experiments show CDP giving a clear advantage over other methods. Additionally, some appendix figures are difficult to interpret because means and confidence intervals overlap heavily (e.g. Fig. 15). This makes it hard to assess some practical differences.

**Questions:**

- How much action diversity is lost as the contraction weight increases? Is there a way to reconcile contraction-type losses with max-entropy RL?


- What is the evidence that diffusion model policies suffer from the same score matching and integration errors that have been observed in e.g. image sampling?

---

> ### Author Response · Authors · 2025-11-19
> **Response to Reviewer rXr6**
>
> We appreciate your insightful feedback, which has indeed sharpened our work. We made every effort to properly address each comment in the order it was presented. Furthermore, we made additional refinements to enhance the paper and conducted further experiments to support our responses. An updated version of the manuscript is released with changes in red.
>
> ```
> 1. Contraction seems to fundamentally contradict a central reason practitioners use diffusion model policies over, e.g. Gaussian policies, namely multimodality. By pulling trajectories together, CDP can collapse valid modes that diffusion policies are meant to capture. It then seems in this setting that Gaussian policies would need to be benchmarked against, or at the very least an explanation provided as to why either (a) multimodality is actually not that important for RL policies, or (b) why the contraction loss does not actually eliminate multimodality when it’s efficient for optimal policies to inhibit this.
> ```
>
> We agree that extreme contraction levels applied globally can be harmful to the multimodal nature of diffusion models, and we are carefully designing the training objective to avoid such scenarios. The contraction loss targets the Jacobian of the score function, which promotes local contraction. This pulls nearby trajectories together to reduce variance and improve robustness, but it does not collapse distinct, well-separated modes. Therefore, preserving multimodality by learning an accurate and diverse score remains a critical objective as long as over-regularization is avoided and the diffusion loss remains critical. **We updated Sec. 3.2 to better discuss the delicate balance between contraction and multimodality.**
>
> In addition, **we added Appendix C.6 to highlight the tools we provide to make over-regularization unlikely.** In particular, we explain that we mitigate collapse by using Frobenius-norm loss with a conservative threshold ($\beta$ in the paper and Appendix C.4) and by enforcing contraction primarily at the latest stages of the denoising process (contr_steps in Appendix C.4). However, since we set these parameters conservatively, we only tune the performance using the contraction regularization parameter $\gamma$ and treat it as the only critical hyperparameter of our approach.
>
> These measures pay off in practice, where we do not observe a measurable collapse over a range of regularization coefficients: CDP improves or matches baselines on most D4RL and Robomimic (compared to DBC) tasks (Tables 1–2), with especially clear gains in low-data regimes (Fig. 5) and on real hardware (Fig. 6). Mode collapse will indeed affect the performance of these tasks, as some (like Antmaze and Kitchen tasks) require multimodal action representation for the best performance. Finally, despite theoretical differences, prior work [6] (in particular Section 4, Theorem 7) provides additional evidence that contraction can improve the process without damaging the flexibility of the diffusion process.
>
> Regarding comparison to Gaussian policies, it is established in the literature that standard (non-contractive) diffusion policies outperform Gaussian policies across benchmarks [1, 2, 3, 4]. We showcase improvement over standard diffusion with our method across the same benchmarks (Table 1-2, Fig. 5). To address your concern more concretely, for imitation learning, **the BC method in Table 2 is already using a GMM policy**, and the results are not comparable to CDP, or diffusion based methods for that matter. **We made this clear in the paper.** For RL, we could easily swap the diffusion policy in one of our baselines, IDQL, with a Gaussian one, resulting in a similar implementation as the original IQL method [5]. **We performed the same experiment as other benchmarks and added these results to Table 1, observing that the IQL method with Gaussian policy only outperforms IDQL (and CDP) in rare cases.** Currently we ran for **5 seeds** to get the results faster for your attention, and we will add more seeds as soon as we have the results.
>
> ```
> 2. There is no direct evidence that the spectral condition for contraction actually holds during training and sampling. Plots of eigenvalues through various time conditioning would give future readers a better idea of the effect of the contraction loss, and what parts of the diffusion model become more strongly contractive.
> ```
>
> Having plots of eigenvalues indeed provides a better insight into enforcing contraction during training. We have already been tracking the largest eigenvalue for debugging purposes and **now added the corresponding plots for some environments to the Appendix F.2 in Figure 13.** Please note that a negative largest eigenvalue means all eigenvalues are negative, and hence the process is contractive. Figure 14 also plots the contraction loss as well as the average reward, which shows different values for different loss weights during training.

---

> ### Author Response · Authors · 2025-11-19
> **Response to Reviewer rXr6 -- Continued**
>
> ```
> 3. As Jacobian computations are typically expensive, a computation time analysis here seems relevant.
> ```
> It is true that our paper could provide a more comprehensive runtime comparison than what is currently shown in Figure 11 of Appendix C.5, which focuses only on the cost of power iterations versus standard Jacobian computation. In response, **we have updated Tables 1 and 2 in our work to include a time comparison against standard diffusion policies.** Please note that our method is built on top of EDP and DBC methods (and DP with the new results), so comparing the average time only makes sense in these cases (eg, DBC-DiT vs. CDP-DiT). The new results indicate that our method incurs a reasonable **overhead of roughly 9% for imitation learning tasks (compared to DBC-DiT) and 14% for reinforcement learning (compared to EDP). **
>
> ```
> 4. While some environments show CDP giving a non-negligible boost, none of the robomimic experiments show CDP giving a clear advantage over other methods...
> ```
> The observation about robomimic is accurate. However, CDP for imitation learning is implemented on top of the DBC architecture, which is known to be less effective than DP in general [3]. This is largely because DP uses a longer prediction horizon and more expressive conditioning networks (we briefly discuss this in Sec. 4.2, Q1). Thus, in Table 2, the most relevant comparison is between CDP and its DBC-based counterparts (DBC-MLP and DBC-DiT), where CDP consistently yields a clear improvement on most tasks and narrows the gap to DP.
>
> To further strengthen these results and in line with your suggestion, **we ran additional Robomimic experiments with CDP implemented directly on the DP architecture through a simple change of diffusion model in our codebase, and added the results to Table 2.** These experiments show an overall advantage over the DBC-based implementations, particularly for more complex vision tasks. Note that we ran for **5 seeds presently**, hence the slightly higher deviations, to get the results faster, but will add 5 more seeds in the coming days.
>
> Lastly, the appendix figures (e.g., Figs. 14 and 16) plot error bands (showing best and worst performance) and not confidence intervals. **We updated the captions** to reflect this.
>
> ```
> Q1. How much action diversity is lost as the contraction weight increases? Is there a way to reconcile contraction-type losses with max-entropy RL?
> ```
> About the loss of action diversity, we talked about why our method will not cause a loss of multimodality in a previous response (the first concern). In sum, contraction pulls nearby trajectories closer by operating on the sensitivity of the reverse ODE via the largest eigenvalue of the symmetric score-Jacobian. This is beneficial to reduce solver and score errors while still keeping the model capable of capturing multi-modal and complex behaviors.
>
> Moreover, max-entropy formulations typically add an entropy bonus on $\pi_\theta(\cdot|s)$ to encourage broad action distributions and push the exploration to find new modalities or valuable actions [like SAC, 7]. Contraction aims to limit numerical/sensitivity error by locally penalizing the score Jacobian in the diffusion sampler. Concretely, one can maximize the maximum-entropy objective while adding our penalty. This requires no architectural changes and uses the same loss components we already define.
>
> ```
> Q2. What is the evidence that diffusion model policies suffer from the same score matching and integration errors that have been observed in e.g. image sampling?
> ```
>
> Policy sampling in both image generation and action diffusion is the same reverse diffusion process: the reverse ODE/Jacobian depends linearly on the score Jacobian $J_{\epsilon_\theta}$, so any score error naturally propagates into the action flow. Likewise, discrete samplers/ODE solvers then add step-size integration error that can accumulate across denoising steps. We state this explicitly and derive $J_{F_\theta}=f(t)I+h(t)J_{\epsilon_\theta}$, which shows the way the score errors can affect the quality of actions. Empirically, we observe this in the outlier samples visible in the toy experiment in Figure 3, and by the general reduction in performance in Figs. 5, 15, 16 when lowering the number of data points, which leads to a less accurate score function. Figs. 17 and 18 expand even more on the toy experiments provided in the paper for better insight.
>
> Most importantly, unlike images, the sequential decision-making nature of robotics tasks makes these small errors consequential. Even small sampling inaccuracies translate to inaccurate actions that cause out-of-distribution states, which give rise to even more out-of-distribution actions in a vicious cycle. Our initial motivation comes from observing these failure scenarios while examining the training process for diffusion policies for toy experiments.

---

> ### Author Response · Authors · 2025-11-19
> **Response to Reviewer rXr6 -- References**
>
> **References**
>
>
> [1] Chi C, Xu Z, Feng S, Cousineau E, Du Y, Burchfiel B, Tedrake R, Song S. Diffusion policy: Visuomotor policy learning via action diffusion. The International Journal of Robotics Research. 2025 Sep;44(10-11):1684-704.
>
> [2] Ze Y, Zhang G, Zhang K, Hu C, Wang M, Xu H. 3D Diffusion Policy: Generalizable Visuomotor Policy Learning via Simple 3D Representations. In 2nd Workshop on Dexterous Manipulation: Design, Perception, and Control (RSS).
>
> [3] Dong Z, Yuan Y, Hao J, Ni F, Ma Y, Li P, Zheng Y. Cleandiffuser: An easy-to-use modularized library for diffusion models in decision making. Advances in Neural Information Processing Systems. 2024 Dec 16;37:86899-926.
>
> [4] Wang Z, Hunt JJ, Zhou M. Diffusion Policies as an Expressive Policy Class for Offline Reinforcement Learning. In The Eleventh International Conference on Learning Representations.
>
> [5] Kostrikov I, Nair A, Levine S. Offline Reinforcement Learning with Implicit Q-Learning. In International Conference on Learning Representations, 2022.
>
> [6] Tang W, Zhao H. Contractive Diffusion Probabilistic Models. CoRR. 2024 Jan 1.
>
> [7] Haarnoja T, Zhou A, Abbeel P, Levine S. Soft actor-critic: Off-policy maximum entropy deep reinforcement learning with a stochastic actor. InInternational conference on machine learning 2018 Jul 3 (pp. 1861-1870). Pmlr.

---

> > ### Comment · Reviewer_rXr6 · 2025-11-21
> >
> > Thank you for such a detailed rebuttal! And apologies for the point I missed in the paper.
> >
> > I have read over the changes, and all of my concerns have been addressed. I don't think computation time necessarily has to be best in class, the point from me was that the transparency would help practitioners understand when to use if (if they can computationally afford it or not), and the method seems generally reasonable compared to baselines.
> >
> > To me, this work's role is to establish that (a) contraction in general is a helpful property to minimize for diffusion / flow-matching policies when balanced appropriately, and (b) the proposed approach, the most natural way to promote contraction, can indeed strike that balance. After the rebuttal, this paper does a good job presenting the idea and giving a good picture of what exactly it does to diffusion model policies and what its purpose is.
> >
> > I am raising my score and I now believe this will have a positive impact on the research community (conditional on promised changes, e.g. more seeds for eval). I saw the synthetic example, but perhaps the paper would further benefit from a multimodality plot from an actual RL instantiation of the paper; something like a maze env. (e.g. in [DDiffPG](https://arxiv.org/pdf/2406.00681)) or UMAP / t-sne projected scatter plots of higher-dimensional robotic actions (e.g. in [QSM](https://arxiv.org/abs/2312.11752)), to give some more evidence that the multimodality balance is reached. Not necessary, but perhaps helpful.

---

> ### Author Response · Authors · 2025-11-21
>
> We appreciate the prompt and indeed positive and encouraging reply. We are happy to be able to address your concern, and will make sure to look into better ways to demonstrate the preserved multimodality as you suggested.
>
> Thank you again for helping us improve our work!
>
> P.S. We also added the additional experiment seeds as promised.

---

### Official Review · Reviewer_xLAA · 2025-11-04

**Soundness:** 3
**Presentation:** 3
**Contribution:** 3
**Rating:** 6
**Confidence:** 3

**Summary:**

This paper proposes a novel diffusion policy called contractive diffusion policy. Comprehensive numerical experiments are conducted.

**Strengths:**

1. The approach has strong theoretical foundations. The authors clearly provide conditions for contractive diffusion policy by leveraging the results in ODE theory and diffusion models.
2. The numerical results are very comprehensive. The paper presents results across multiple benchmarks, showing the effectiveness of the proposed approach.
3. Writing is easy to follow.

**Weaknesses:**

Although I am familiar with diffusion models theory, I am not familiar with diffusion policy benchmarks. I only have one major concern:
1. In Tang and Zhao (2024), the contractive condition excludes the application of usual diffusion model such as VP SDE. Do you face the same issue in your setup? If yes, what diffusion process are used and how to you compare with usual DDPM-based sampling process?

**Questions:**

The paper is clear to me. No further questions.

---

> ### Author Response · Authors · 2025-11-19
> **Response to Reviewer xLAA**
>
> We appreciate your feedback and thank you for recognizing our theoretical contribution and thorough experiments. We have responded to your feedback and have also implemented revisions and experiments to improve the paper based on your comments,  with changes in red.
> ```
> 1. In Tang and Zhao (2024), the contractive condition excludes the application of usual diffusion model such as VP SDE. Do you face the same issue in your setup? If yes, what diffusion process are used and how to you compare with usual DDPM-based sampling process?
> ```
>
> Unlike CDPMs [Tang and Zhao (2024), 1], our method naturally accommodates VP-SDE formulations, as illustrated in the proof of Theorem 3.1 in App. D.2. Since VP-SDE architectures are widely used in robot learning applications, we primarily adopt VP-SDE scheduling in our experiments. To make this connection more explicit, **we have updated App. D.2 with an additional discussion of VE-SDE formulations, outlining how our core theorem can be extended to VE schedulings as well.**
>
> To address the second concern, we instantiate the diffusion ODE sampler for efficiency and use VP-SDE scheduling. The diffusion ODE formulation also aligns more naturally with our theoretical development, allowing us to more clearly articulate the benefits of contraction in this setting. Nevertheless, our derivations in App. E show that the same Jacobian regularization can be applied to DDPM [3] and DDIM [4] as well. In particular, App. E focuses on DDPM and DDIM sampling processes and demonstrates how the relationship between the score Jacobian and the Jacobian of the reverse process can still be derived in these cases.
>
> Lastly, we note that in our codebase, switching between solvers is straightforward: one can simply change the configuration key “solver” to any of the supported options. Although we expose DDPM-based sampling as an option, we did not initially include DDPM-based comparisons, since our main focus is on SDE/ODE-based diffusion sampling. However, given your concern and the ease of changing solvers in our implementation, **we conducted additional experiments in Figure 12, Appendix E, comparing DDPM, DDIM, and ODE solvers on the Franka Kitchen [2] tasks, observing comparable performance.**
>
> We hope that our response has sufficiently addressed your concerns. We are here in case there are more questions or concerns.
>
> **References**
>
> [1] Tang W, Zhao H. Contractive Diffusion Probabilistic Models. CoRR. 2024 Jan 1.
>
> [2] Fu J, Kumar A, Nachum O, Tucker G, Levine S. D4RL: Datasets for deep data-driven reinforcement learning. arXiv preprint arXiv:2004.07219. 2020 Apr 15.
>
> [3] Ho J, Jain A, Abbeel P. Denoising diffusion probabilistic models. Advances in neural information processing systems. 2020;33:6840-51.
>
> [4] Song J, Meng C, Ermon S. Denoising diffusion implicit models. arXiv preprint arXiv:2010.02502. 2020 Oct 6.

---

### Official Review · Reviewer_5Yr9 · 2025-11-05

**Soundness:** 3
**Presentation:** 3
**Contribution:** 2
**Rating:** 4
**Confidence:** 4

**Summary:**

This paper proposes to improve diffusion policy in the continuous control setup by inducing contractive behaviors. While the iterative sampling process of diffusion models encourages diverse outputs, it also hinders action generation in fields that require accurate signals (e.g., robotic control). The method introduces a penalty loss term on the score Jacobian to enhance robustness in the sampling process, as it effectively reduces action variance. The authors show that this offline learning method can improve the diffusion policy baseline on several robotic control problems, especially in the limited training data regime.

**Strengths:**

+ The method is simple and intuitive. The results look promising (especially the real-world experiments).
+ The authors provide an in-depth theoretical analysis of their proposed method for inducing the contraction of diffusion models.
+ The work is positioned well in a detailed discussion of related work.

**Weaknesses:**

+ I am not fully convinced of the idea of unwanted action variance. Granted, excessive noise in the generated actions can lead to failure in continuous control. But the level of variance that is beneficial varies a lot from task to task, as pointed out in the appendix. It would be better if the authors could come up with a more principled tuning method (e.g., a self-adaptive coefficient). In particular, for tasks that require dynamic control (say, soft-body manipulation), it might be hard to justify the contraction-based regularization.
+ I suspect that it might be harder for the contraction-regularized diffusion model to do transfer learning or further online fine-tuning. Basically, with the reduced action variance, it might be harder to adapt to various action distributions in a post-training manner. While the promise of a large diffusion policy for action generation lies in a foundation model + quick post-training fine-tuning paradigm, I wonder if the proposed approach might hinder this.
+ The Jacobian (or in general Lipschitz) regularization approach has been popularized by a series of works in DNN generalization [1,2,3]. Please add them to the related work section.

I am willing to raise my score after seeing the authors’ response.

[1] Information-theoretic local minima characterization and regularization, ICML 2020

[2] Sharpness-aware minimization for efficiently improving generalization, ICLR 2021

[3] Understanding Gradient Regularization in Deep Learning: Efficient Finite-Difference Computation and Implicit Bias, ICML 2023

**Questions:**

Does the distillation process (e.g., consistency policy) effectively also induce a contraction? As the reduced sampling steps and the consistency distillation loss might also “pull” nearby diffusion flows closer. I suggest the authors study their proposed loss on distilled diffusion policies as well.

---

> ### Author Response · Authors · 2025-11-19
> **Response to Reviewer 5Yr9**
>
> Thank you for your thoughtful and detailed comments and for recognizing the intuitive nature and the theoretical depth of our work. We did our best to respond to your feedback point by point, and have accordingly updated the paper to reflect the comments with changes in red.
> ```
> 1. I am not fully convinced of the idea of unwanted action variance. Granted, excessive noise in the generated actions can lead to failure in continuous control. But the level of variance that is beneficial varies a lot from task to task, as pointed out in the appendix. It would be better if the authors could come up with a more principled tuning method (e.g., a self-adaptive coefficient). In particular, for tasks that require dynamic control (say, soft-body manipulation), it might be difficult to justify the contraction-based regularization.
> ```
> We agree with the reviewer that a fixed contraction weight $\gamma$ is not ideal across different tasks, and we explicitly acknowledge this limitation and its implications in the paper. Intuitively, our goal is to enforce just enough contraction to obtain its benefits, without pushing it so far that action variance and multimodal structure collapse. In the current version of our method, we address this using **a simple hyperparameter sweep over six candidates**, $[0.001, 0.01, 0.1, 1.0, 10, 100]$ (Sec. 4.1), and we find that meaningful improvements already emerge under this naïve, low-overhead tuning procedure. Furthermore, as shown in Table 4 in the appendix, once a particular value of $\gamma$ performs well for a given environment, it tends to remain effective across different tasks within the same environment (e.g., real-world tasks). This suggests that even a modest tuning on one environment with a small candidate set can improve performance over standard diffusion policies.
>
> At a deeper level, the main difficulty is that the true performance of diffusion policies with different variance levels manifests only at test (or validation) time, making it difficult to assess the real-world performance of diffusion policies before deployment. In other terms, determining the true range of beneficial action variance is extremely hard and often conducted at test time. Plus, the appropriate scale for our contraction regularization term can depend on factors such as the structure of the action distribution, demonstration quality, sample size, and other task-dependent variations that are difficult to predict in advance.
>
> Regarding more dynamic control tasks, we would be happy to run additional experiments if the reviewer has specific suggestions and if the explanation above remains unconvincing. Nonetheless, because contraction reduces sampling sensitivity without substantially altering action magnitude, bandwidth, or multimodality, we expect the benefits to extend to more dynamic control settings as well. Exploring soft body manipulation is indeed an interesting direction for future work.
>
> ```
> 2. I suspect that it might be harder for the contraction-regularized diffusion model to do transfer learning or further online fine-tuning. Basically, with the reduced action variance, it might be harder to adapt to various action distributions in a post-training manner. While the promise of a large diffusion policy for action generation lies in a foundation model + quick post-training fine-tuning paradigm, I wonder if the proposed approach might hinder this.
> ```
>
> We certainly think that post-training large foundational models with action diffusion head [2, 3] is an intriguing application of contraction to investigate. Theoretically, we believe that contraction can still provide benefits for post-training a large model with an action diffusion head, since the contraction regularization term is additive and can be properly adjusted or annealed during post-training. We already provide some features in our code base that could make contraction more adapted to post-training scenarios. For instance, we can apply contraction only in late denoising steps (“contr_steps” parameter in the codebase and Appendix. C.4), which we use precisely when handling complex distributions. Alternatively, one could change the contraction threshold in the Frobenius loss if needed (“contr_thr” in the code and Appendix. C.4). Such properties can be effective for training larger models with action diffusion heads.
>
> **We added Appendix. C.6 to better explain these extended options when training larger models.**
> In addition, we considered conducting experiments with large foundation models, but the **computational and setup complexities make it nearly impossible to have these results on short notice.** If the reviewer believes that such comparisons are useful in addition to our explanation above, we can try to train a small foundation model with action diffusion [3] for robomimic tasks to better support our argument. However, these models are around the same size as the transformer-based models we use in the experiments (~2X larger).

---

> ### Author Response · Authors · 2025-11-19
> **Response to Reviewer 5Yr9 -- Continued**
>
> ```
> 3. The Jacobian (or in general Lipschitz) regularization approach has been popularized by a series of works in DNN generalization [1,2,3]. Please add them to the related work section.
> ```
> Thank you for noticing this. The literature of Lipschitz regularization is indeed relevant to what we do (essentially a form of Lipschitz regularization). **We added the mentioned papers by extending our literature review in App. A.4.**
>
>
>
> ```
> Question. Does the distillation process (e.g., consistency policy) effectively also induce a contraction? As the reduced sampling steps and the consistency distillation loss might also “pull” nearby diffusion flows closer. I suggest the authors study their proposed loss on distilled diffusion policies as well.
> ```
>
> Thank you for bringing up this point. Consistency-style distillation can reduce sensitivity if the loss is strongly enforced by collapsing actions along the same diffusion trajectories. However, this does not address the problem of solver and score-matching errors, and the work is primarily focused on enhanced computation time of visuomotor policies by aligning solutions across time steps and shrinking teacher-student discrepancies. As a result, in the original "consistency policy" paper [5], we observe that the performance is **on par with diffusion policies** and rarely surpasses the performance of standard diffusion policies (Figure IV. for instance), unlike our approach, which effectively improves the overall performance (Table 1, 2). Our method is grounded in the theory of differential equations to provide finer control over the behavior of all denoising steps. **We will expand our literature to cover consistency policies and their relation to our work.**
>
>
> We again appreciate your help in improving our work. Please let us know if you have more questions/concerns about the work or our responses.
>
>
> **References**
>
> [1] Haarnoja T, Zhou A, Abbeel P, Levine S. Soft Actor-Critic: Off-Policy Maximum Entropy Deep Reinforcement Learning with a Stochastic Actor.
>
> [2] Wen J, Zhu Y, Li J, Zhu M, Tang Z, Wu K, Xu Z, Liu N, Cheng R, Shen C, Peng Y. TinyVLA: Towards fast, data-efficient vision-language-action models for robotic manipulation. IEEE Robotics and Automation Letters. 2025 Feb 24.
>
> [3] Intelligence P, Black K, Brown N, Darpinian J, Dhabalia K, Driess D, Esmail A, Equi M, Finn C, Fusai N, Galliker MY. $\pi_ {0.5} $: a Vision-Language-Action Model with Open-World Generalization. arXiv preprint arXiv:2504.16054. 2025 Apr 22.
>
> [4] Shukor M, Aubakirova D, Capuano F, Kooijmans P, Palma S, Zouitine A, Aractingi M, Pascal C, Russi M, Marafioti A, Alibert S. Smolvla: A vision-language-action model for affordable and efficient robotics. arXiv preprint arXiv:2506.01844. 2025 Jun 2.
>
> [5] Prasad A, Lin K, Wu J, Zhou L, Bohg J. Consistency policy: Accelerated visuomotor policies via consistency distillation. arXiv preprint arXiv:2405.07503. 2024 May 13.

---

### Author Response · Authors · 2025-11-19
**Revised Manuscript Uploaded**

The manuscript is updated to reflect the changes we outlined in the first round of responses. We will have a minor update only to add the additional experiment seeds to better support some responses.

---

> ### Author Response · Authors · 2025-11-24
> **Minor Manuscript Updates**
>
> As promised, we updated the manuscript to include the additional experiment seeds and a few minor edits to enhance the clarity of our figures and tables.

---

### Meta-Review · Area_Chair_pYKH · 2026-01-01

**Summary:**

This paper introduces a new approach to improving diffusion policies by inducing contractive behavior in diffusion models. All reviewers agree that the theoretical analysis is solid. Furthermore, many reviewers appreciate that the idea is simple and does not change the main architecture of diffusion models, making the method straightforward to apply.

While several reviewers initially raised concerns regarding the potential collapse of multimodality in RL policies, the authors mostly  addressed this in the rebuttal. They provided clarifications and additional experiments to demonstrate that the method avoids such collapse. Their rebuttal was successfully, reflected by the fact that reviewer rXr6 increased the score from 2 to 6. After reviewing the paper, the comments, and the author responses, I believe this work makes a clear contribution to the RL community. Therefore, I recommend acceptance.

**Reviewer Concerns:**

Reviewer 5Yr9: This reviewer questioned the benefit of variance reduction, suggesting it might hinder dynamic control tasks and future fine-tuning, and requested the inclusion of missing literature on Lipschitz regularization. From my point of view, the authors only partially address these points. They explained that "determining the true range of beneficial action variance is extremely hard and often conducted at test time", and due to the time constraint, they were not able to run large foundation models for transfer learning and online fine-tuning.

Reviewer xLAA: This reviewer only raised one concern and I believe it has been addressed.

Reviewer rXr6: This reviewer initially suggested rejection, by questioning that the method may lead to the collapse of multimodality and computational cost of Jacobian calculations can be high. After detailed explanation and more experiments, these concerns were resolved, and the reviewer subsequently increased their score from 2 to 6.

Reviewer T4PE: This reviewer expressed concerns regarding the computational overhead of the Jacobian-vector products, the potential for contraction to undermine the multi-modal strengths of diffusion models, and the high sensitivity of the regularization hyperparameter across different environments. By providing the computational time, conducting further experiments, and adding more clarifications, I think the authors have largely mitigated these concerns.

**Reviewer Scores:**

Reviewer rXr6 increased the score from 2 to 6. Reviewer 5Yr9 may increase the score if the discussion can be continued.

---

### Decision · Program_Chairs · 2026-01-26

Accept (Poster)